## OPEN

# Single-particle cryo-EM structures from iDPC–STEM at near-atomic resolution

Ivan Lazić [1] ✉, Maarten Wirix[1], Max Leo Leidl [2,3,4,5], Felix de Haas[1], Daniel Mann [2,5], Maximilian Beckers[2,6], Evgeniya V. Pechnikova[1], Knut Müller-Caspary[3,4], Ricardo Egoavil[1], Eric G. T. Bosch[1] and Carsten Sachse [2,5,7] ✉

In electron cryomicroscopy (cryo-EM), molecular images of vitrified biological samples are obtained by conventional transmission microscopy (CTEM) using large underfocuses and subsequently computationally combined into a high-resolution three-dimensional structure. Here, we apply scanning transmission electron microscopy (STEM) using the integrated differential phase contrast mode also known as iDPC–STEM to two cryo-EM test specimens, keyhole limpet hemocyanin (KLH) and tobacco mosaic virus (TMV). The micrographs show complete contrast transfer to high resolution and enable the cryo-EM structure determination for KLH at 6.5 Å resolution, as well as for TMV at 3.5 Å resolution using single-particle reconstruction methods, which share identical features with maps obtained by CTEM of a previously acquired same-sized TMV data set. These data show that STEM imaging in general, and in particular the iDPC–STEM approach, can be applied to vitrified single-particle specimens to determine near-atomic resolution cryo-EM structures of biological macromolecules.

Scanning transmission electron microscopy (STEM) is a well-established methodology in characterizing materials at micro, nano and atomic scales[1–3]. STEM methods and a derivate known as ptychography were shown to image dose-resistant specimens reaching resolutions better than 0.5 Å (refs. [4,5]), the latter becoming the method of choice for obtaining the highest possible spatial resolutions[6]. Among these STEM imaging modalities is integrated differential phase contrast–STEM (iDPC–STEM)[7,8], which has been routinely applied to a variety of specimens such as GaN, $NdGaO_3$-$La_{0.67}Sr_{0.33}MnO_3$, Ni–YSZ interfaces and $Bi_2Sr_2CaCu_2O_{8+\delta}$ superconductors[9–12]. In the same manner, specimens such as metal hydrides were successfully visualized at subatomic resolution, including heavy elements alongside light elements such as hydrogen[13]. Moreover, iDPC–STEM was demonstrated to successfully image different crystalline as well as amorphous materials including beam-sensitive ones such as zeolites[14–16]. One of these samples included an individual aromatic hydrocarbon molecule trapped within a porous framework structure[17]. Other investigated materials are known as metal-organic frameworks that can only be imaged using electron doses smaller than $50\,e^-/Å^2$ before damaging the structure[18]. By using iDPC–STEM with a low-dose exposure of as little as $40\,e^-/Å^2$, a resolution of 1.8 Å was obtained successfully from a single micrograph for such materials[19]. Recently, large biological sections, including thick ones (~500 nm), have also been imaged[20]. For these dose-sensitive low-contrast specimens, iDPC–STEM enables direct interpretation of the image without the need of defocusing and a subsequent contrast-transfer function (CTF) correction of the images[21].

One of the early STEM applications to freeze-dried biological samples was the molecular mass determination by annular dark field (ADF) scattering, as the number of atoms is directly related to the scattering intensity[22]. More recently, STEM tomography has been applied to thick vitrified cells[23,24]. Although low in resolution, it was shown that image contrast can be obtained even from micrometer-thick samples mainly by inelastically scattered electrons[25]. Furthermore, cryo-STEM was applied to single-particle specimens of Fe or Zn-loaded ferritin to precisely visualize and locate metals within the protein cage[26]. More recently, micrographs of rotavirus and HIV-1 virus-like particles were imaged using ptychography reporting the principal applicability to biological specimens albeit at low resolution[27]. Thus far, complete three-dimensional (3D) single-particle cryo-EM structures of biological macromolecules at close-to-atomic resolution, have not been determined by images of the STEM technique.

Electron cryomicroscopy (cryo-EM) has become a very successful structural biology technique that commonly produces near-atomic resolution 3D structures of ice-embedded biological macromolecules. Micrographs are obtained by conventional transmission electron microscopy (CTEM) at low fluences of $20–100\,e^-/Å^2$ and imaged at micrometer-defocuses to enhance low-resolution contrast of the macromolecules. When images of proteins assembled in regular two-dimensional (2D) or helical arrays were available, they could be successfully determined at high resolution to enable atomic model building decades ago[28–30]. Due to the advent of improved hardware and software, also known as the resolution revolution[31], single-particle structures are now routinely resolved at near-atomic resolution. Thus, hundreds to thousands of micrographs are acquired at varying underfocus to increase the contrast of the ice-embedded macromolecules. Subsequently, elaborate image processing work-flows are used to process the molecular

[1]Materials and Structural Analysis Division, Thermo Fisher Scientific, Eindhoven, Netherlands. [2]Ernst Ruska-Centre for Microscopy and Spectroscopy with Electrons (ER-C-3): Structural Biology, Jülich, Germany. [3]Department of Chemistry and Centre for NanoScience, Ludwig-Maximilians-University Munich, Munich, Germany. [4]Ernst Ruska-Centre for Microscopy and Spectroscopy with Electrons (ER-C-1): Physics of Nanoscale Systems, Jülich, Germany. [5]Institute for Biological Information Processing (IBI-6): Cellular Structural Biology, Jülich, Germany. [6]Structural and Computational Biology Unit, European Molecular Biology Laboratory (EMBL), Heidelberg, Germany. [7]Department of Biology, Heinrich Heine University, Düsseldorf, Germany. ✉e-mail: ivan.lazic@thermofisher.com; c.sachse@fz-juelich.de

projections, perform CTF correction and determine their orientation parameters to constructively merge often more than 10,000 molecular views in a final 3D image reconstruction[32,33]. At present, structures are commonly resolved better than 4.0 Å, suitable for atomic model building. To benchmark the performance of the cryo-EM method, the obtained resolutions of various test specimens such as tobacco mosaic virus (TMV) were improved over time to better than 2.0 Å (ref. [34]). So far, the highest resolutions of 1.2 Å were accomplished by using another common test specimen known as apoferritin[35,36].

Due to the reported benefits of STEM approaches for a large range of different materials, we wanted to explore whether STEM imaging can be applied to vitrified biological samples and produce high-resolution images. Using iDPC–STEM imaging of KLH, we here report the successful subnanometer single-particle cryo-EM structure at 6.5 Å resolution. More systematically, using iDPC–STEM imaging of TMV, we further demonstrate the successful single-particle based helical reconstruction at 3.5 Å resolution using an electron beam of 4.0 mrad convergence semiangle (CSA). The resulting cryo-EM map matches the expected features of previously analyzed CTEM TMV data sets at the given resolution. The obtained quality of the iDPC–STEM map exceeds those obtained with CTEM approaches recorded in 2015 using second-generation direct electron detection (DED) cameras[37]. Our study shows that iDPC–STEM imaging can be successfully applied to cryo-EM single-particle-based structure determination and elucidate biological structures at near-atomic resolution.

## Results

**Imaging of cryo-samples by iDPC–STEM.** To successfully image vitrified samples by STEM, several additional aspects need to be considered in comparison with CTEM defocus-based imaging. First, instead of flood-beam illumination of a large field of view (FOV), the convergent beam moves over the specimen in regular steps and illuminates the sample spot by spot to scan the region of interest (Fig. 1a,b). For each beam position of the scanning process, the resulting signal is recorded in the far field behind the sample using a center-of-mass detector, here approximated by means of a four-quadrant detector[7,8,38,39]. Second, whereas in CTEM the optical resolution limit is caused by an inserted aperture and lens aberrations, in STEM the CSA of the focused beam controls the apparent resolution and the depth of focus (Fig. 1c). When aberrations are present, they will only affect the CTF shape but not limit the resolution[8]. Third, unlike CTEM imaging methods that produce additional phase contrast by defocusing the specimen, STEM techniques such as ADF–STEM and iDPC–STEM reach the highest contrast of the image in focus. Therefore, focus and maximized contrast was obtained by examining the flatness of the convergence beam electron diffraction (CBED) pattern of the beam (Fig. 1d,e). For a typical experiment with an opening angle of 2.0 mrad CSA (Fig. 1f), a probe spot of 4.9 Å effective diameter of the beam intensity moves

over the sample line by line scanning every 2.4 Å spot resulting in an overlap of 50% exposed area. Each spot exposure lasts a dwell time of 4 µs. The acquired spot signals are combined pixel by pixel to generate a complete 4,096 × 4,096 pixels micrograph over a total of 68 s time yielding a FOV of 983 nm. In analogy to existing cryo-EM low-dose acquisition protocols, we avoid additional exposures of the molecules and applied the focusing procedure next to the area of interest on the carbon foil.

To experimentally verify the optical resolution of the STEM approach, we analyzed a standard 50-nm-thick sample of gold deposited on a carbon film and confirmed the presence of the 2.3 Å gold ring in the power spectrum of the images obtained at 4.5 mrad CSA and electron dose of 350 e−/Å² (Fig. 1g and Supplementary Fig. 1). The first diffraction ring of the gold lattice was straightforward to detect and the obtained resolution was found very close to the theoretical resolution limit of 2.2 Å for the chosen CSA (Table 1 and Supplementary Table 1). In addition, we noted the absence of any typical Thon rings (Supplementary Fig. 2a,b) that are common for CTEM imaging. Instead, we observed a fourfold star pattern at the origin of the power spectrum, reflecting the CTF of iDPC–STEM. In the ideal case when center of mass is directly detected, the resulting CTF and power spectrum will be rotationally symmetric and decay toward higher frequencies. When a four-quadrant detector is used, the resulting 2D CTF has a fourfold pattern in addition to the resolution decay. The fourfold CTF pattern dominated the low spatial frequencies only and has a minor effect on the visual appearance of the image[8,9] (Supplementary Fig. 2c,d). In comparison with an oscillating CTF of a CTEM image acquired in underfocus, the signal in iDPC–STEM further decays almost linearly toward higher spatial frequencies reaching zero at the theoretical limit of the given CSA (Supplementary Fig. 2e–g). Using the described imaging setup, an iDPC–STEM micrograph of ice-embedded keyhole limpet hemocyanin (KLH) was recorded at a CSA of 2.0 mrad (Fig. 1h), which exhibits for instance strong contrast of ice contamination as a prominent low-frequency transferred feature in addition to KLH particles. For consistent interpretation of iDPC–STEM images, we applied a high-pass filter at 251 Å full-width at half-maximum referring to them as preprocessed iDPC–STEM images. These iDPC–STEM micrographs resemble inverted CTEM images in appearance as CTEM images contain very few low spatial frequencies (Fig. 1i). In conclusion, iDPC–STEM micrographs provide the complete projections of the sample including low and high frequencies[8,40], which in appearance are comparable to in-focus CTEM images using a phase plate[41].

**3D structure of vitrified KLH from iDPC–STEM micrographs.** To quantitatively analyze larger numbers of KLH images, we collected a total of 760 micrographs at a CSA of 2.0 mrad. After extracting a total of 9,150 particles and performing routine 2D classification procedures[42], we identified a series of averaged characteristic side, top and tilted side views of KLH compatible with D5 symmetry

**Fig. 1 | iDPC–STEM imaging setup, acquisition and micrographs. a**, Schematic model of the probe formation system, sample and four-quadrant detector (left). The red line illustrates the average electron path of the beam, the red dot indicates the center of mass (COM) of the intensity at the detector. The inset (center) corresponds to a longitudinal cross-section through the aberration-free probe. The z scale along the beam is highly compressed with respect to the xy plane for visualization (not drawn to scale). Cartoon model (right) of the probe indicates optical properties: CSA, beam-waist width/resolution and length/depth. **b**, Illustration of the acquisition process: focusing is performed on the carbon foil next to the region of interest (left), before the scanning process starts (center and right). **c**, Probe shape, that is depth of focus, for different CSAs with respect to the sample thickness (not drawn to scale in the z-direction). **d**, Focusing using CBED pattern (inset), out of focus (**d**) and in focus (**e**). **f**, Illustration of the scanning process. The zoomed inset shows a scanning step of 2.4 Å to ensure maximal resolution using a 2.0 mrad CSA beam of 4.9 Å (Table 1). **g**, Experimental iDPC–STEM micrograph (4,096 × 4,096 pixels) of a gold-on-carbon sample (one out of ten acquired) confirming an optical resolution of 2.3 Å (inset top right) acquired at 300 kV, CSA of 4.5 mrad, electron dose of 350 e−/Å² and 0.76 Å pixel size, scale bar is 50 nm. **h**, Raw iDPC–STEM micrograph (4,096 × 4,096 pixels) of vitrified KLH acquired at 300 kV, CSA of 2.0 mrad, with electron dose of 40 e−/Å² and 1.5 Å pixel size, scale bar is 50 nm (one out of ~1,000 acquired). Note the strong low-frequency contrast of the ice contamination. **i**, Corresponding Gaussian high-passed filtered (preprocessed) iDPC–STEM micrograph matching the appearance of typical CTEM images of **h** with additional atomic models of two different KLH projections drawn to scale, scale bar is 50 nm.

(Fig. 2a). To improve resolution for a 3D reconstruction, we collected additional 687 micrographs using a larger CSA of 3.5 mrad, extracted 18,597 particles and determined the structure of KLH with D5 symmetry imposed at 6.5/6.8 Å (0.143 and false discovery rate–Fourier shell correlation (FDR–FSC) criterion) (Fig. 2b,c), which is beyond the latest reported resolution[43]. We docked the available structure Protein Data Bank (PDB) ID 4BED into the density and found good agreement with the expected secondary

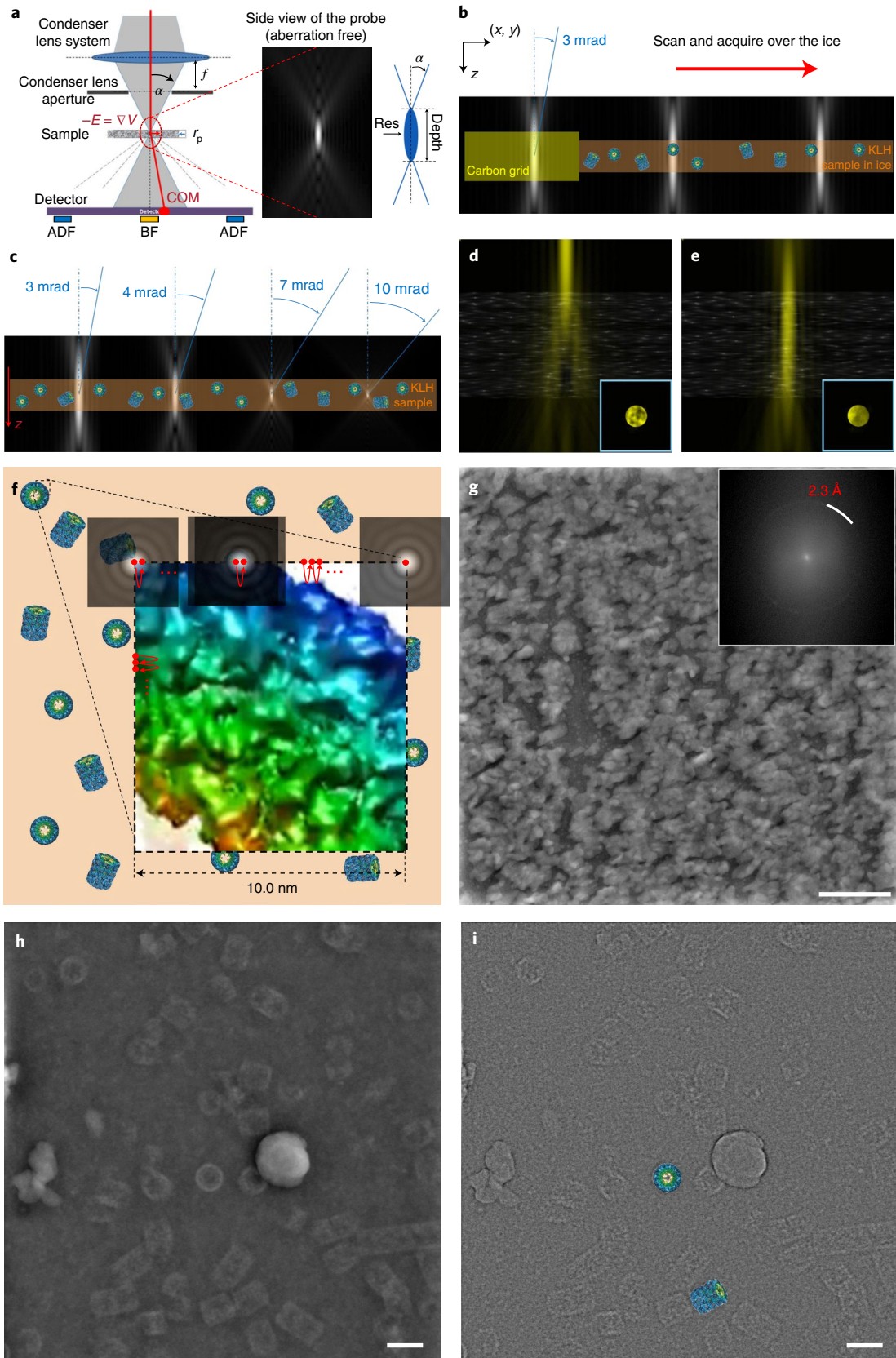

**Table 1 | Critical STEM imaging parameters for cryo-EM image acquisition at 300 keV electrons (wavelength $\lambda = 1.969$ pm)**

| CSA $\alpha$ (mrad) | Experimental STEM resolution based on Au measurement (Å) | Maximum theoretical STEM resolution $\lambda/(2\alpha)$ (Å) | Required pixel size for maximum STEM resolution (Å) | Depth of focus $2\lambda/\alpha^2$ (nm) |
|---|---|---|---|---|
| 2.0 | 5.1 | 4.9 | 2.4 | 985 |
| 3.0 | 3.5 | 3.3 | 1.6 | 438 |
| 3.5 | 3.0 | 2.9 | 1.4 | 321 |
| 4.0 | 2.6 | 2.5 | 1.2 | 246 |
| 4.5 | 2.3 | 2.2 | 1.1 | 194 |

structural elements at the determined resolution (Fig. 2d). These data show that iDPC–STEM micrographs of a single-particle specimen can be used to determine the subnanometer molecular structure of KLH.

Further improvements in resolution of the 3D structure of KLH would require a data set size orders of magnitude larger. In the light of this consideration, we turned to the molecular specimen TMV, containing a high number of asymmetric units per unit length, which is ideally suited for efficient structural averaging using a smaller number of micrographs. The helical organization of TMV and resulting image repeats are used to assess information transfer of the micrographs. Initial images were acquired with a low CSA of 2.0 mrad at a pixel size of 1.70 Å (Fig. 2e) and STEM image size of 4,096 × 4,096 pixels. The corresponding FOV of 696 × 696 nm² is comparable to CTEM cryo-image acquisitions that record a part of a micrometer carbon hole. The iDPC–STEM micrograph reveals the presence of TMV rods densely packed in rafts close to the edge of the carbon hole. As the contrast results directly from the electrostatic potential, TMV's protein density is white and not inverted as in CTEM. Closer inspection of the fine structure shows the distances of the 23 Å as well as the 11.5 Å repeats along the helical rod. Moreover, we acquired more images of densely formed TMV rafts using a CSA beam of 4.0 mrad (Fig. 2f). The corresponding Fourier transform shows a continuous information transfer, characteristic TMV layer lines and the absence of any CTF zero crossings (Fig. 2g). The noticeable vertical streak along the meridian of the image Fourier transform is a well-known scanning effect in STEM[44]. In the Fourier transform, we were able to detect high-resolution information up to 1/7.7 and 1/4.6 Å⁻¹, which indicated further potential of cryo-iDPC–STEM imaging at high resolution.

**3D structures of vitrified TMV at different CSAs.** Next, we set out to establish imaging conditions of the TMV sample to obtain optimal high-resolution 3D image reconstructions. As two defocused-based CTEM reference data sets, we used TMV images deposited with accession IDs EMPIAR-10305 (ref. [34]) and EMPIAR-10021 (ref. [45]). Subsequently, we systematically compared iDPC–STEM data sets taken with beam CSAs of 2.0, 3.0, 3.5, 4.0 and 4.5 mrad respecting critical parameters given in Table 1, with the CTEM reference cryo-micrographs. For the detailed comparison of all data sets, we analyzed a typical micrograph including the corresponding Fourier transform (Fig. 3). Under all imaging conditions, iDPC–STEM images show strong low-frequency contrast in comparison with the defocused cryo-image. The respective Fourier transforms show the expected 1/23 and 1/11.5 Å⁻¹, first- and second-order layer lines, whose intensities are primarily determined by the number of TMV rods present in the FOV. For more quantitative analyses, we extracted 1,300–2,200 helical segments from several micrographs and computed a power spectrum average,

displayed as a one-dimensional (1D) helical profile. Comparing the overall slope of the 1D helical profile revealed that the profiles derived from the iDPC–STEM images decay faster than from the defocused CTEM micrographs. The relative ratio of the second over first order layer-line profile peaks increases from 0.2 to 0.5 with a CSA beam increase from 2.0 to 4.5 mrad, respectively, which indicates improved signal transfer of the higher resolution layer line at higher CSA. The corresponding ratio of the latest defocused CTEM data set (EMPIAR-10305) is 1.9, surpassing the 0.5 of the 3.5, 4.0 and 4.5 mrad STEM data sets.

Finally, using the individual CTEM and iDPC–STEM micrographs, we processed approximately the same number of helical segments from the different data sets using the typical single-particle helical reconstruction workflow of RELION[46] (Table 2). It should be noted, however, that for iDPC–STEM in the absence of defocusing, we used neither CTF determination nor any CTF correction options. Subsequently, we performed the 3D structure refinements using the 2.0, 3.0, 3.5, 4.0 and 4.5 mrad beam-CSA data sets and determined the TMV structures at 6.3, 4.3, 3.9, 3.5 and 3.7 Å resolution, respectively, based on the 0.143 FSC cutoff criterion[47,48] (Supplementary Fig. 3a). The trend of near-atomic resolution values of high CSA acquisitions at 3.5, 4.0 and 4.5 mrad could also be confirmed independently by a mask-less FDR–FSC determination approach[49] at 3.4, 3.2 and 3.3 Å, respectively (Supplementary Fig. 3b). The latest CTEM data subset from 2019 (EMPIAR-10305) went to 2.2 Å. When compared with the best iDPC–STEM map (4.0 mrad CSA) at 3.5 Å resolution, the earlier 2015 CTEM data set (EMPIAR-10021) went to slightly poorer resolution of 3.7 Å. Local resolution assessment corroborates the better map quality of the 4.0 mrad CSA iDPC–STEM structure in comparison with the 2015 CTEM structure albeit worse than the 2019 CTEM structure (Supplementary Fig. 3c).

To validate the numerical resolution assessment, we inspected the CTEM and iDPC–STEM cryo-EM density maps in more detail (Fig. 4a,b). In the presence of a previously refined atomic model (PDB 4UDV)[45], we find that secondary structures are well discernible for the 2.0 mrad map at 6.3 Å resolution (Fig. 4c). The density of the α-helical pitch can be well recognized for the 3.0 mrad map at 4.3 Å resolution (Fig. 4d). In addition, the maps also show clear density of the RNA and for ribose moieties that are tightly packed between the subunits. Qualitatively, the molecular features that are discernible in 3.5, 4.0 and 4.5 mrad maps at near-atomic resolution constitute larger side chains, that is aromatic and positively charged ones such as F35, R41, W52, F62, F87, R113 and R122, in addition to a well-defined polypeptide backbone (Fig. 4e–g). In analogy to the 2015 CTEM map (EMPIAR-10021) that received the same total electron dose of 35 e⁻/Å², density for negatively charged side chains D77, E106, D115, D116 and E131 is largely absent. The displayed cryo-EM densities were sharpened based on the Guinier plot analysis of the 3D reconstruction. The corresponding Guinier $B$ factors were determined to 492, 155, 105, 132 and 126 Å² for the 2.0, 3.0, 3.5, 4.0 and 4.5 mrad maps, respectively. The smaller Guinier $B$ factors correlate with the generally improved resolution and image quality for the 3.5–4.5 mrad data sets. $B$ factor estimations that assess the data set quality as a whole involve the analysis of the particle number as a function of the obtained resolution[47]. Using particle subsets of CTEM (EMPIAR-10021), CTEM (EMPIAR-10305), 3.5 mrad iDPC–STEM KLH and 4.0 mrad iDPC–STEM TMV micrographs, we determined $B$ factors by logarithmic regression to 147, 62, 437 and 93 Å², respectively (Supplementary Fig. 3d). A global $B$ factor of 93 Å² suggests that the present iDPC–STEM micrographs are of better quality than the CTEM acquisition from 2015 recorded using a Falcon II DED[42], and of worse quality than the 2019 CTEM acquisition by a K2 DED CTEM acquisition in 2019 (ref. [34]). Together, the cryo-EM density analysis of vitrified TMV confirms that in-focus iDPC–STEM images can be reliably used to resolve the detailed

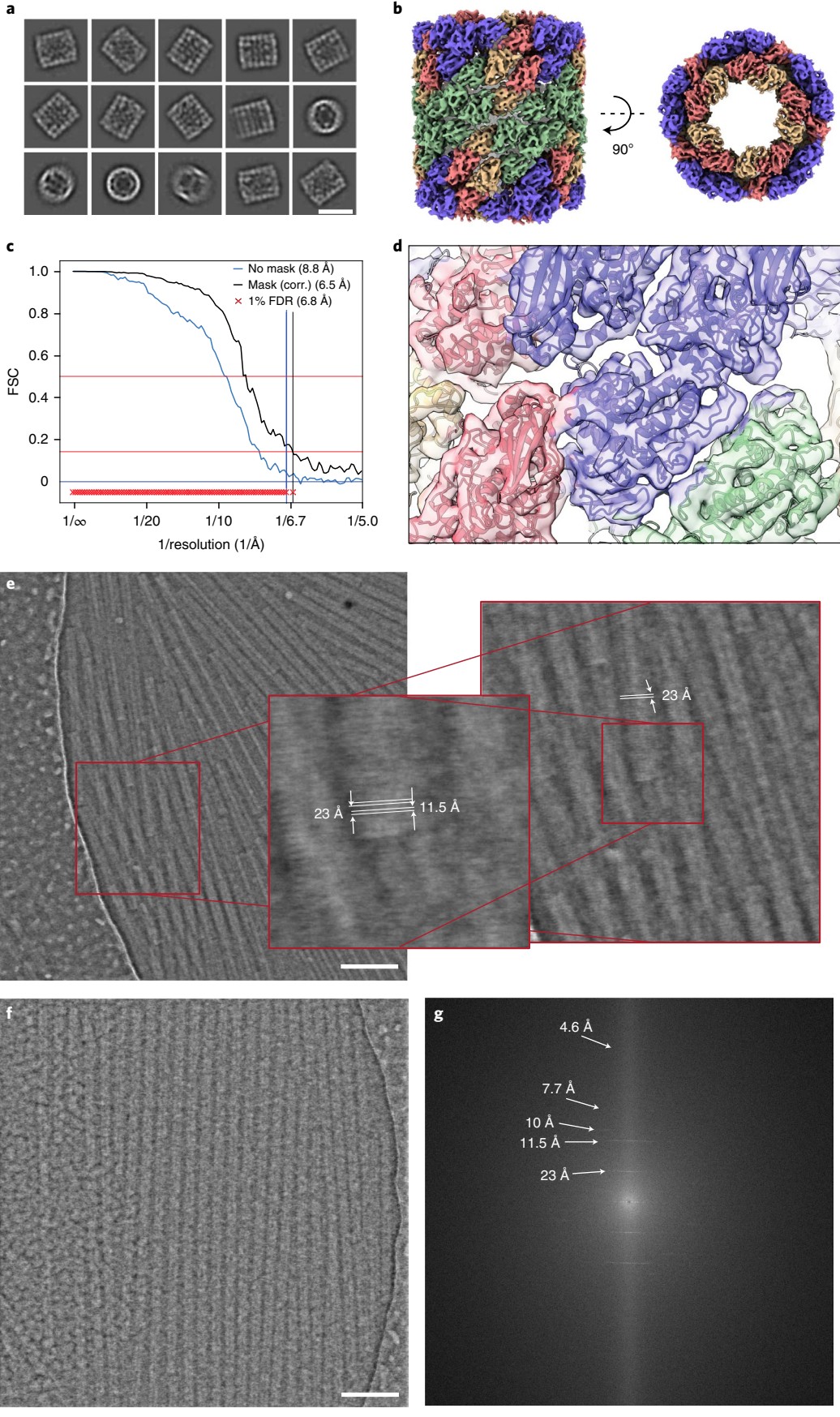

**Fig. 2 | Two biological specimens imaged by iDPC–STEM.** Top, cryo-iDPC–STEM single-particle reconstruction of KLH. Experimental conditions: voltage 300 kV, CSA 3.5 mrad, pixel size 1.2 Å, image size 4,096 × 4,096 pixels, total electron dose 35 e⁻/Å². Scale bar, 300 Å. **a**, Representative 2D classes at CSA of 2.0 mrad (9,150 particles). **b**, Reconstructed 3D structure at CSA of 3.5 mrad with D5 symmetry in side and top views (18,597 particles). **c**, FSC indicates a nominal resolution of 6.5 and 6.8 Å using 0.143 and FDR–FSC cutoff, respectively. **d**, PDB 4BED docked inside the cryo-EM density (color code as in **b**). Bottom, typical cryo-iDPC–STEM micrographs of TMV acquired at 2.0 and 4.0 mrad CSA. Experimental conditions: voltage 300 kV, pixel size 1.70 Å, image size 4,096 × 4,096 pixels, total electron dose 35 e⁻/Å². **e**, Preprocessed iDPC–STEM image at 2.0 mrad CSA shows TMV particles in ice with carbon foil on the left, scale bar is 100 nm (one out of 20 micrographs). Zoomed in regions indicate the helical 11.5 and 23 Å repeats along the rod in real space. **f,g**, Preprocessed iDPC–STEM image at 4.0 mrad CSA beam of a TMV raft, scale bar is 100 nm (one out of 20 micrographs) (**f**) and the corresponding power spectrum. Layer lines show high-frequency information up to 1/4.6 Å⁻¹ (white arrows) (**g**).

protein structures of ice-embedded biological macromolecules to near-atomic resolution.

The estimated resolutions of the determined iDPC cryo-EM structures confirm the basic optical considerations that increasing CSA leads to higher spatial resolution. To evaluate additional limiting parameters of the imaging setup, we simulated iDPC–STEM images of a hemoglobin molecule embedded in vitreous ice. First, on increasing the CSA beam, we observed a decrease in the signal-to-noise ratio (SNR) of the particle (Supplementary Fig. 4a). When the same number of electrons is used, smaller CSA beams will produce higher SNRs due to the reduced solid-angle volume of the electron-wave behind the sample. Therefore, for initial micrograph assessment, smaller beam CSAs are recommended for boosting SNR. The apparent SNR can also be matched by increasing the electron dose at the expense of radiation damage effects. Second, at high CSAs and large pixel sizes, aliasing may obstruct the image. To illustrate this effect in simulated images, we used a large CSA beam of 10 mrad at a pixel size larger than the resolution cutoff imposed by the CSA. Under these conditions, the discernible particle contrast weakened because higher resolution information is folded back to low frequencies (Supplementary Fig. 4b). To avoid aliasing, we paid attention that the scan interval, that is pixel size, fully samples the optical resolution provided by the beam CSA (Table 1). For example, the highest resolution iDPC–STEM TMV map at 3.5 Å resolution was achieved with the 4.0 mrad CSA beam and a pixel size of 0.98 Å using an electron dose of 35 e⁻/Å².

## Discussion

Using iDPC–STEM, we imaged plunge-frozen KLH and TMV as biological test specimens to assess the image quality and analyzed the resulting 3D cryo-EM density maps. We demonstrated that iDPC–STEM produces high-contrast in-focus micrographs of ice-embedded molecular KLH particles and the regular helical image features of TMV. When we subjected multiple micrographs to the single-particle reconstruction workflow, we determined the subnanometer KLH structure at 6.5 Å and several near-atomic resolution TMV structures using different iDPC–STEM imaging conditions down to 3.5 Å. By using the appropriate combination of CSA beam and associated imaging parameters we demonstrated the capability of obtaining resolution within the expected theoretical limits (Tables 1 and 2). These data establish that iDPC–STEM imaging of cryo-vitrified biological samples generates micrographs of sufficiently high quality suitable for near-atomic resolution single-particle cryo-EM structure determination.

One of the apparent features of the recorded iDPC–STEM cryo-micrographs is that they show continuously transferred contrast over the complete frequency band. This favorable property is specific to the iDPC–STEM approach. For instance, annular bright-field (ABF)-STEM suffers from CTF shortcomings similar to CTEM. Due to the reciprocity theorem ABF-STEM also requires defocus to generate contrast and to enhance the low-frequency transfer[50]. In addition, only part of the scattered electrons is collected by the detector and used to form an image. For ADF–STEM, imaging can be performed in focus resulting in an overall positive CTF. However, the total number of electrons collected in the dark field is several orders of magnitude smaller than in the bright field, making ADF detection very dose-inefficient[14]. Moreover, for ADF–STEM the imaged object corresponds to the square of the electrostatic potential[8,50], causing the light elements to practically disappear when imaged next to heavier ones. Based on these considerations, standard STEM techniques do not exhibit sufficient capabilities to be exploited for low-dose imaging of radiation-sensitive materials. Conversely, iDPC–STEM uses all signal-relevant electrons, suppresses noise in the integration step[7,8] and has a favorable CTF devoid of any contrast reversal and CTF zero crossings. The imaged object is directly proportional to the electrostatic potential field of the sample[8,50]. When imaging gold particles deposited on a carbon film using iDPC–STEM with a 4.5 mrad CSA beam, we resolved the details to 2.3 Å resolution, confirmed by the first gold-lattice diffraction ring in the power spectrum of the image (Fig. 1). Due to the detector architecture of four quadrants, we also observed a fourfold shaped 2D CTF. This CTF pattern can be compensated by a 2D CTF correction[9] as the theory is well understood[7,8]. However, we did not find this necessary in our analysis, as KLH particles or helical TMV segments with random orientations within the ice layer plane are averaged. The power spectra of the recorded iDPC–STEM cryo-micrographs, as expected, did not show any Thon rings that are common for defocused-based CTEM imaging. The resulting high-resolution signal transfer in iDPC–STEM is, therefore, very well suited for near-atomic cryo-EM structure determination.

The performance of iDPC–STEM depends on experimental parameters that must be controlled during imaging. The beam CSA critically determines the maximum possible resolution of the STEM image (Table 1). This resolution is exactly related to the beam width and the beam depth of focus, also referred to as depth resolution or beam-waist length, according to theoretical considerations[50–52]. When increasing the CSA and the probe is aberration-free, higher resolution is obtained while the beam depth of focus becomes smaller. Consequently, when the sample thickness exceeds the decreasing beam-waist length, the images represent optical depth sections rather than projections[50,53]. We estimate that the sample regions imaged here have thicknesses of 18 to 54 nm

**Fig. 3 | Comparison of TMV cryo-images taken by conventional TEM (CTEM) and iDPC–STEM.** Micrograph inset (CTEM and preprocessed iDPC–STEM) of vitrified TMV (left, scale bar 100 nm), corresponding power spectrum (center left), four in-plane rotated TMV segments (center right shows two left column iDPC–STEM images, two right column high-pass filtered iDPC–STEM images to match the appearance of CTEM images, scale bar 10 nm) and corresponding layer-line profile of added power spectra segments. In the layer-line profile, first- and second-order layer lines give rise to peaks I and II at 1/23 and 1/11.5 Å⁻¹, respectively. The ratio of II/I is given in the upper right corner. The orange line is averaged from the corresponding data sets of around 2,000 segments and the blue line is obtained from averaged segments of a single micrograph. **a–g**, CTEM (defocused-based) cryo-images EMPIAR-10305 (**a**) and EMPIAR-10021 (**b**) followed by 2.0 mrad (**c**), 3.0 mrad (**d**), 3.5 mrad (**e**), 4.0 mrad (**f**) and 4.5 mrad (**g**) CSA beam iDPC–STEM cryo-images.

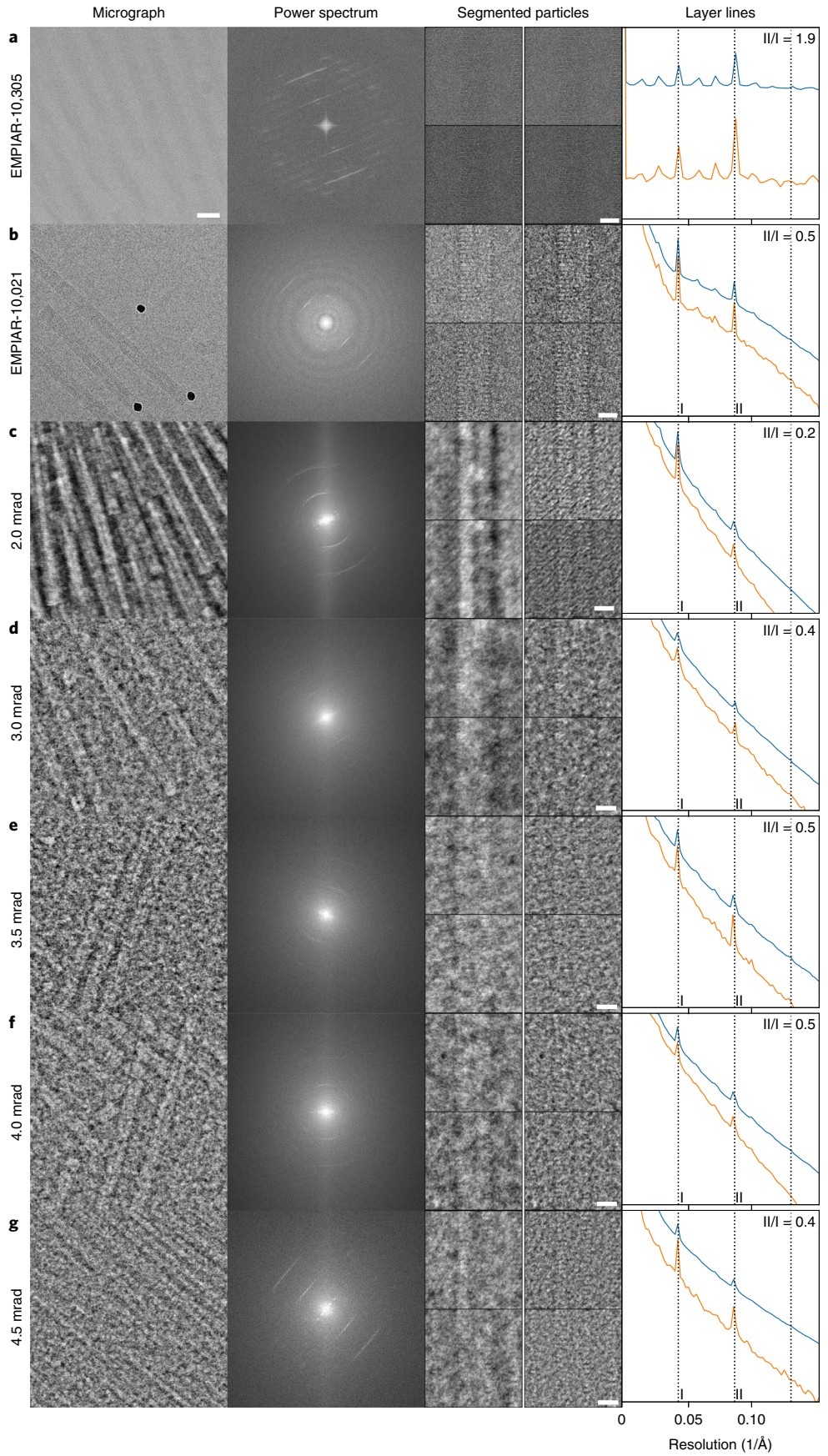

**Table 2 | Summary of 3D reconstruction results of reference CTEM and described cryo-iDPC–STEM data sets**

| | CTEM/ EMPIAR-10021 | CTEM/ EMPIAR-10305 | iDPC–STEM 2.0 mrad | iDPC–STEM 3.0 mrad | iDPC–STEM 3.5 mrad | iDPC–STEM 4.0 mrad | iDPC–STEM 4.5 mrad |
|---|---|---|---|---|---|---|---|
| Pixel size (Å) | 1.062 | 0.638 | 1.705 | 1.23 | 0.98 | 0.98 | 0.75 |
| Number of micrographs | 28 | 62 | 20 | 13 | 15 | 20 | 28 |
| Number of segments | 2,068 | 3,270 | 2,073 | 1,392 | 1,330 | 2,224 | 1,678 |
| Number of asymmetric units | 53,768 | 75,210 | 87,066 | 41,760 | 31,920 | 75,616 | 57,052 |
| $B$ factor (Å²) | 126 | 38 | 492 | 155 | 105 | 132 | 126 |
| Maximum resolution target (Table 1) (Å) | NA | NA | 5.1 | 3.5 | 3.0 | 2.6 | 2.3 |
| Resolution expected range (Supplementary Table 1) (Å) | NA | NA | 7.4–5.6 | 5.0–3.8 | 4.3–3.3 | 3.8–2.9 | 3.3–2.5 |
| Resolution FSC (0.143, ref. [47]) (Å) | 3.8 | 2.2 | 6.3 | 4.3 | 3.9 | 3.5 | 3.7 |
| Resolution FSC (FDR[49]) (Å) | 3.6 | 2.1 | 5.9 | 4.1 | 3.4 | 3.2 | 3.3 |

corresponding to a diameter of a single TMV rod and three TMV rods, respectively. When typical ice thicknesses of 10–60 nm, common for plunge-frozen specimens, are considered, only the beam CSAs larger than 9 mrad at 300 kV will not be sufficient for obtaining complete projections of the sample. With the maximum beam CSA of 4.5 mrad used in this study resulting in a depth of focus of 194 nm, complete projections are produced in all cases. This property of STEM imaging is substantially different to CTEM acquisitions when different particle z-positions within the ice layer give rise to different projections due to the associated change in CTF. For higher CSAs, for example larger than 7.0 mrad, in addition to decreasing the depth of focus, the spherical aberration of the probe deteriorates the image due to the introduction of CTF zero crossings[52] (Supplementary Table 1). This property justifies the usage of probe aberration-correctors when higher spatial resolutions are targeted. The discussed beam relationship of resolution versus depth of focus also opens up principal possibilities for high-resolution imaging of thicker cryo-frozen samples using optical sectioning STEM[40,54,55].

To compare CTEM and STEM acquisitions, we emphasize their fundamental difference with respect to electron delivery and energy deposition. Unlike with flood-beam illumination in CTEM where electrons are delivered everywhere at once, the scanning mode deposits electrons sequentially one pixel position at a time. A potential benefit of the STEM approach may be that the energy transferred to the sample can spread and dissipate toward nonilluminated areas, presumably weakening the impact and the damage at the exposed spot. Due to the overlapping geometry of the scan, that is effective beam size is larger than a pixel, the sample spots are exposed multiple times and, consequently, the total dose per pixel is accumulating in STEM. In contrast, CTEM exposed areas are illuminated once and, in some cases, damage-free maps can even be extrapolated at zero electron exposure[56]. In analogy to STEM, spot scanning of 100 nm beams was implemented in CTEM and was shown to mitigate beam-induced motion and to improve contrast for vitrified specimens[57]. The STEM approach offers additional opportunities for evaluating different electron delivery strategies by, for example, changing the scan grid order. This scanning scheme may give rise to reduced radiation damage effects when compared with typical flood-beam illumination approaches. A series of studies investigating the damage mechanisms in STEM has been reported in 2D materials using alternative scan patterns, which indicate that they may further reduce beam damage[58–60]. The typical CTEM exposure causes beam-induced motion and leads to ice-patch or particle movements and ultimately to image blur, which is now commonly compensated by motion correction[61]. In this paper, motion correction

was also applied to the 2015 and 2019 TMV CTEM reference data sets. For STEM, the locally induced beam motion building up throughout the scan may result in image distortions and anisotropic magnifications in the reconstructed micrograph. Nevertheless, without the employment of any correction strategies, the present iDPC–STEM experiments show that a simple grid scanning scheme can be used to generate near-atomic resolution structures of TMV.

Reprocessing of a TMV CTEM data sets (EMPIAR-10021, ref. [45]; EMPIAR-10305, ref. [34]), limited to approximately 50,000 asymmetric units and resulted in a map of 3.7 and 2.2 Å resolution. The 4.0 mrad iDPC–STEM map at 3.5 Å resolution showed the expected cryo-EM density features at the given resolution (Fig. 4 and Supplementary Fig. 3). Associated map sharpening $B$ factors determined by Guinier analysis of both the CTEM and 4.0 mrad iDPC–STEM maps are at around 130 Å². For poorer resolutions of TMV data sets taken on film, higher $B$ factors of 240 and 280 Å² were reported[30,62] whereas for TMV data sets taken on the Falcon III DED and K2 cameras, lower $B$ factors of 100 and 40 Å² have been determined[34,63] (Supplementary Table 2). Further quantitative comparison of global $B$ factors by estimating the resolution as a function of TMV particle subsets places the 4.0 mrad iDPC–STEM data set in quality between the 2015 and 2019 CTEM data sets. Therefore, it is remarkable that iDPC–STEM, in combination with a four-quadrant detector without any motion correction, shows an improved performance and $B$ factor in comparison with CTEM imaging from 2015 using a second generation DED camera including motion correction. The map features indicate that the final iDPC–STEM map exhibits very similar radiation damage effects on the negatively charged side chains as other CTEM maps[45,64,65]. The iDPC–STEM spot-scanning approach appears not to suffer critically from beam-induced movement that used to be one of the critical resolution-deteriorating issues before the introduction of the movie mode in DED cameras[66].

Working out appropriate iDPC–STEM parameters for imaging vitrified specimens was straightforward as TMV required relatively few micrographs for quantitative image analysis and 3D reconstruction. The scanning procedures used for the micrograph acquisition took approximately 20–80 seconds for the 5.0 and the 2.0 mrad CSA beams, respectively (Supplementary Table 3). This way, approximately 60 TMV micrographs were collected in each CSA session in the absence of any automation. Collecting large data sets up to 300 micrographs per hour, common for typical CTEM single-particle acquisitions[67], would not be possible in this manner. Automation of the STEM acquisition procedures, however, will be straightforward to employ in the future. Detectors used for iDPC–STEM allow scan speeds of two orders of magnitude faster than the ones used in this work, which will ultimately reduce acquisition times to below

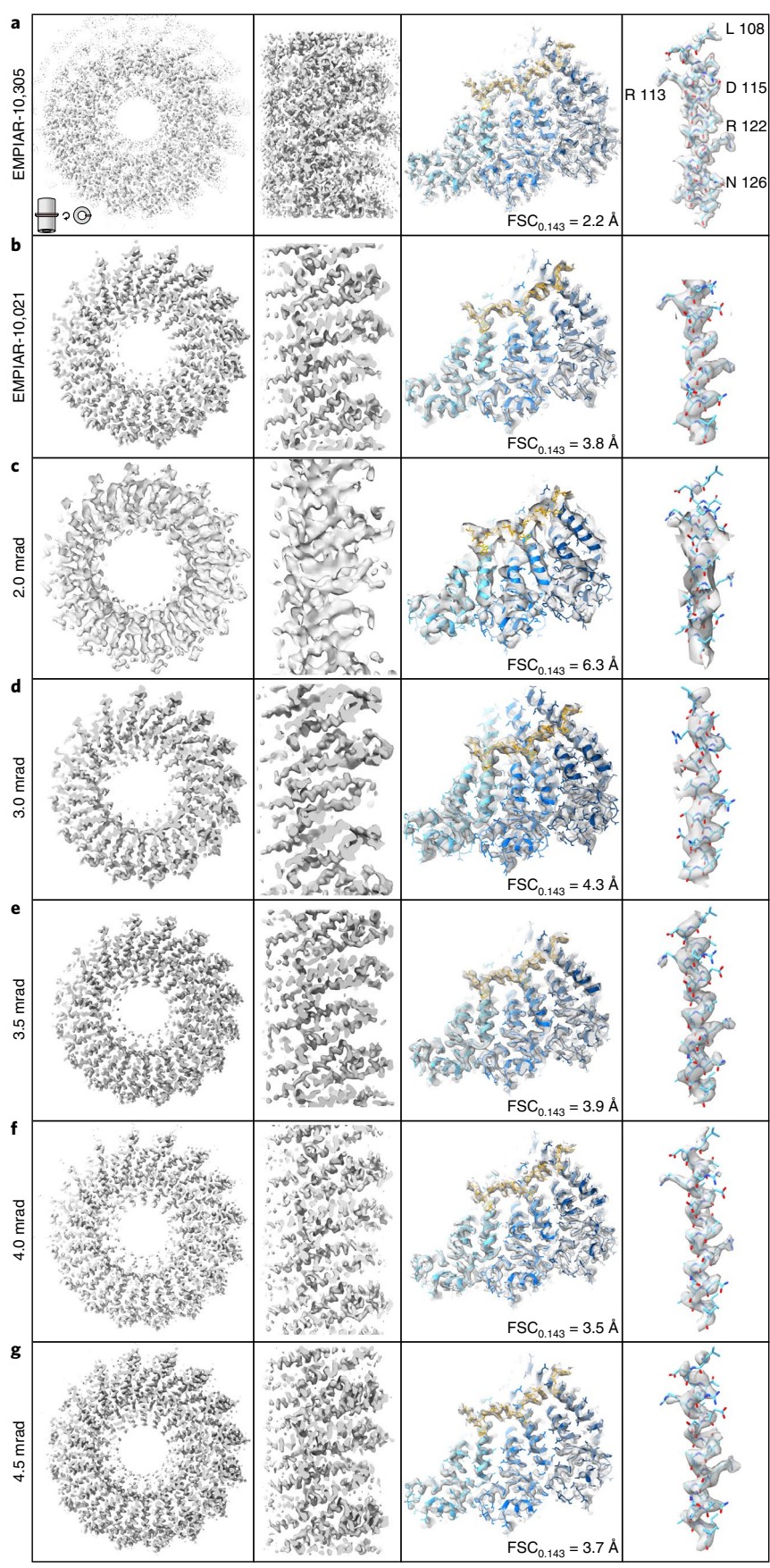

**Fig. 4 | Comparison of cryo-EM density maps with fitted trimer of TMV coat protein PDB 6SAG and 4UDV reconstructed from conventional TEM (CTEM) EMPIAR-10305 and EMPIAR-10021, respectively and iDPC–STEM images using different CSA beams.** Top slab view of TMV map (left), enlarged side slab view of multiple TMV subunit densities (center left), three-subunit top view (center right) and detailed side-chain density of α-helix (108–130). **a–g**, Cryo-EM densities from CTEM EMPIAR-10305 (**a**), CTEM EMPIAR-10021 (**b**) and from 2.0 mrad (**c**), 3.0 mrad (**d**), 3.5 mrad (**e**), 4.0 mrad (**f**) and 4.5 mrad (**g**) beam-CSA iDPC–STEM images. The EMPIAR-10305/EMPIAR-10021 CTEM maps have a resolution of 2.2/3.8 Å and the 2.0/3.0 mrad iDPC–STEM maps at 6.5/4.3 Å resolution, respectively. The 3.5, 4.0 and 4.5 mrad CSA cryo-iDPC–STEM maps share the same molecular features with the CTEM 3.8 Å near-atomic resolution map. The 4.0 mrad CSA iDPC–STEM data set gives the best resolution of 3.5 Å of compared iDPC–STEM data sets. All 3–4 Å resolution maps lack density for negatively charged residues (for example, D115).

one second. This way, when hardware improvement supported by software-based automation is further implemented, it will become possible to image equally large data sets of single-particle specimens in similar time frames as in CTEM. Additionally, recording individual frames in movie mode will reduce the effects of beam-induced motion and further improve the image quality of cryo-iDPC–STEM.

In this study, we demonstrated the principal applicability of STEM methods to cryo-vitrified biological specimens for 3D structure determination. Advanced STEM methods or derivates have been used in materials science for resolving sub-50 pm spatial detail[4,5]. Some of these methods, such as iDPC–STEM[7,8], have been shown to be very dose efficient and, therefore, have been applied successfully to dose-sensitive specimens[18,19]. Images can be obtained with highest contrast in focus without the typical contrast inversions at higher frequencies known from CTEM. One of the further benefits of iDPC–STEM over CTEM is the improvement in low-resolution contrast when using smaller CSAs to target intermediate resolutions. Largest benefits can be expected when no or little averaging of cryo-images is possible. In the case of biological vitrified samples, this feature may be particularly useful for the visualization of complex environments such as cellular lysates and biological cells[20]. In CTEM, the thickness of the specimen imposes a substantial limitation on such samples as contrast blurs due to incoherency and inelastic scattering. The problem remains critical even when in-focus images are taken with a phase plate. For STEM, the contrast will not be significantly deteriorated by inelastic scattering, also at lower voltages the image remains sharp and shows full contrast even when the object thickness becomes comparable to the inelastic mean free path[68]. Dynamic focusing across a scan field of a tilted specimen is possible and could provide benefits in single-particle analysis to overcome preferred particle orientations. Once iDPC–STEM imaging can be applied to tilt series of thicker biological specimens, cryo-electron tomographic reconstructions will benefit from the complete contrast transfer in STEM images devoid of any oscillating CTF while still preserving high-resolution information. Moreover, alternative STEM acquisition schemes[60] and micrograph reconstruction approaches[27], when further developed, should also prove beneficial for imaging biological samples. The STEM approach opens up additional opportunities for correlative sample characterization, for example spectroscopic mapping by energy dispersive X-ray spectroscopy or electron energy loss spectroscopy to analyze the element composition of a specimen. The results presented here, using iDPC–STEM to image vitrified KLH and TMV specimens, reveal that STEM approaches should be further explored for improved high-resolution cryo-EM structure determination.

## Online content

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

## Methods

**Specimen preparation.** Quantifoil grids were rendered hydrophilic using a glow discharge device (GloQube Plus, Quorum) for a time of 45 s in air before specimen application to the grid. An aliquot of 3.5 µl stock solution of TMV at a concentration of approximately 90 mg ml$^{-1}$ or 2.5 µM was applied onto a 200 mesh Quantifoil grid with regular R2/2 holey carbon support. The excess of the applied droplet was blotted using a Vitrobot Mark IV (Thermo Fisher Scientific MSD). Due to the high concentration, optimal results for obtaining thin ice across the 2 µm holes were achieved using a high blot force of +10 and a duration for blotting of 10 s, before plunge freezing. Grids were prepared at 4 °C and 100% humidity. To maximize occupancy of holes with rafts of TMV rods, application of TMV at high concentrations was critical. After flash freezing, grids were stored in grid boxes in liquid nitrogen for subsequent mounting into autogrids of the Krios G4 Cryo-Autoloader. For the KLH samples, cryo-grids were prepared as described above. For KLH, however, a 10 mg ml$^{-1}$ or 2.5 µM solution was plunge frozen using a blot force of −10 and 6 s blotting time.

**STEM.** STEM imaging was conducted using a Thermo Fisher Scientific Titan Krios G4, operated at 300 kV. The column was equipped with a standard high-brightness field emission gun (X-FEG), three-condenser lens system, C-TWIN objective lens with wide-gap pole piece (11 mm and $C_s = 2.7$ mm), Panther segmented STEM detector, a Ceta camera and Falcon 4 DED camera. A combination of different C2 apertures (20 and 50 µm) and C2/C3 lens current ratios were used to create different CSA of the beam. Before STEM imaging, beam shift, beam tilt pivot points and beam tilt in STEM mode were aligned for the different convergence angles. For accurate determination of the COM, a descan alignment was performed in addition. The CSA of the applied beams were measured with high precision using the Au cross-grating and Ceta camera by first recording the radii of the gold diffraction rings for calibration and subsequently measuring the radius of the bright-field disc. The beam current was determined using the fluorescent (flu-) screen, calibrated by a Faraday-cup measurement. The total electron dose (TED) delivered per unit square area is given by

$$\text{TED} = \frac{I \times t_{\text{frame}}}{\text{FOV}^2} = \frac{I \times t_{\text{dwell}}}{\text{PS}^2},$$

where $I$ is the beam current (number of electrons per unit time) while $t_{\text{frame}}$ is total scanning time required to scan the entire FOV frame area. Further, $t_{\text{dwell}}$ is the time that the beam spends per pixel, and PS is the pixel size. Note that $t_{\text{frame}} = n \times t_{\text{dwell}}$ where, $n = N^2$ number of pixels of for example $N \times N$ size image, while $\text{FOV} = N \times \text{PS}$. Therefore, the number of pixels cancels and yields the right-hand side of the equation. The beam current was held at fixed 4 pA while the dwell time was used as a parameter to fine-tune the final total dose of 35 e$^-$/Å$^2$. Low magnification atlas screening, searching for a suitable ice thickness area, was performed in TEM mode with the Falcon 4 DED and the MAPS software. Intermediate magnification ADF–STEM images were collected at ×5,000 magnification to get a STEM overview image and assign it to the TEM atlas. Using the MAPS software, the different magnifications and modes were correlated and aligned. A low magnification atlas (for example a tiled set of images representing most of the grid area), served as a navigation map for finding grid squares that have suitable ice thickness and sufficiently dense rafts of TMV specimen. Smaller 4 × 4 or 5 × 5 tiled atlas maps were created within the grid squares to navigate the stage to the holes containing TMV. Data were acquired and stored at each tile position across the holes using Velox v.3.2 software.

**iDPC–STEM imaging and acquisition.** For iDPC–STEM acquisition, the four-quadrant mode of the Panther STEM detection system (Thermo Fisher Scientific) was used. The detector has a circular layout with a hole in the center and is composed of eight segments where each quarter circle is split into an inner and outer segment (Supplementary Fig. 5a). The central hole has one-sixth inner radius in relationship to the overall detector dimension (Supplementary Fig. 5b). The eight segments are hard wired in such a way that radial separation is not used resulting in four quadrants with read-out capability. The beam size on the detector is set to cover more than two-thirds of the detector diameter such that the central hole radius corresponds to less than one-quarter of the beam size (Supplementary Fig. 5c). The scan grid geometry consists of a simple line by line layout contributing to each pixel of the reconstructed micrograph (Supplementary Fig. 5d). The offset angle between the scanning coordinate frame and the detector coordinate frame, which determines the scan direction, is measured and set initially during the alignment. The scan direction can be chosen freely and is taken into account by providing illumination center of mass vector components for the resulting scanning coordinate frame. This vector determination is critical to perform before the final integration step in iDPC–STEM takes place[7]. Note that only when the rotation angle of the scan direction is zero, the components of the DPC-STEM vector can be simply determined by subtraction of the signal from the opposite quadrants[7,8,38].

For every convergence angle, the camera length, that is the distance from the sample to the detector plane (Supplementary Fig. 6a), was chosen such that bright-field-disk of the beam at the detector covers detector area with a four times larger radius than the radius of the central hole. With an additional ADF detector, simultaneous acquisition of ADF/ABF and iDPC–STEM data is possible. Exemplary micrographs of vitrified densely packed TMV rods confirmed improved contrast and high-resolution information transfer of iDPC–STEM acquisitions (Supplementary Fig. 6b,c). For each beam CSA ≥4.5 mrad, gold rings were used in the power spectrum of STEM (iDPC, Fig. 1g and Supplementary Fig. 2c,d and ADF, Supplementary Fig. 1) to confirm the resolution by imaging a standard cross-grating gold-on-carbon sample. As mentioned above, the total applied electron dose 35 e$^-$/Å$^2$ was used with parameters listed in Supplementary Table 3. Therefore, the typical acquisition time for a 4,096 × 4,096 pixel-sized micrograph was between 13.2 s at 0.75 Å pixel size and 67.9 s at 1.70 Å pixel size. Approximately 10 s rest-time of the stage was applied between acquisitions. Focusing was performed at the carbon film grid next to the area of interest (hole with ice and particles) by judging the flatness of the CBED pattern (Ronchigram). For KLH, approximately 700 to 800 micrographs were acquired for each CSA session. For TMV, typically 50 to 70 micrographs were acquired per CSA session (Table 2). The micrographs with the highest resolving layer lines (approximately one-third) were selected and used for helix segmentation and 3D reconstruction. For data recording, the procedure was followed in MAPS v.3.16 software (Thermo Fisher Scientific) for optimal, accurate and safe stage navigation. The overview maps served to navigate and collect high-resolution iDPC–STEM images preventing multiple exposures.

**Image processing and 3D single-particle reconstruction.** The raw iDPC–STEM micrographs were preprocessed by applying a Gaussian high-pass filter with a full-width at half-maximum of 251 Å and imported without contrast inversion and no CTF information. For KLH a total of 760 and 687 iDPC–STEM micrographs were acquired for CSA of 2.0 and 3.5 mrad, respectively, and subjected to single-particle analysis in CryoSPARC v.3.3.1 (ref. [42]). For the beam CSA of 2.0 mrad data set, 9,750 particles were extracted, subjected to 2D classification and an ab initio 3D model was generated (batch size 1,000). For the CSA 3.5 mrad data set, 500 particles were manually picked to generate classes for template-based particle picking. Subsequently, a total of 120,727 putative particles were extracted with 18,597 particles remaining after 2D classification (batch size 200 with 200 iterations) and subjected to refinement using an initial low-pass filtered map of the CSA 2.0 mrad initial model (60 Å filter cutoff) with imposed D5 symmetry. The 3D reconstruction showed clear secondary structure information with a nominal resolution of 7.7 Å (FSC 0.143) and was followed by nonuniform local refinement[69], including D5 symmetry to yield a final map with a nominal resolution of 6.5 Å (FSC 0.143) and 6.8 Å (FSC–FDR). The 3D reconstruction was sharpened by 522.7 Å$^2$ based on Guinier $B$ factor estimation. PDB 4BED (ref. [43]) was docked inside the EM density using ChimeraX v.1.3 (ref. [70]).

For the processing of TMV micrographs, helical coordinates were interactively picked using EMAN2 (ref. [71]). The in-plane rotated segments were used to calculate the averaged power spectra. The power spectra sums were collapsed in the direction orthogonal to the helical axis into 1D spectra as described[72]. Collapsed power spectra were used to calibrate the pixel size by matching the visible layer lines with the expected layer lines of $\{\frac{n}{22.03} | n = 1, 2, 3, \dots\}$. Further image processing was performed using Relion v.3.1 (ref. [73]). Depending on the data set, one or two rounds of 2D classifications with ten classes were performed. For the following steps, only particles from classes showing high-resolution details of TMV were included. Due to the absence of defocusing in iDPC–STEM and overall positive CTF[8,9], neither CTF determination nor any CTF correction option was used. Smaller subsets of EMPIAR-10305 and EMPIAR-10021 CTEM data were reprocessed according to the standard single-particle helical reconstruction workflow. To ensure comparability with the iDPC–STEM data sets, motion correction was performed and in the case of EMPIAR-10021 limiting the included frames to a total dose of 35 e$^-$/Å$^2$ with the exposure/dose weighting option turned off. To generate a reference for the 3D refinement, 3D classification with one class and a featureless cylinder as a reference was performed. After refinement, a mask was created from one of the half maps, including the central 30 volume percentage in the $z$-direction. Subsequently, postprocessing and local resolution estimation were performed. The half maps were used for FSC calculation including mask deconvolution[48] and resolution taken at the 0.143 threshold[47]. To validate the estimated resolutions with another criterion, the mask-free FDR–FSC approach[49] was applied to the central 30 volume percentage in the $z$-direction. Cryo-EM density analysis was carried out using available TMV coordinates PDB 6SAG (ref. [34]) for EMPIAR-10305 and 4UDV (ref. [45]) for EMPIAR-10021 and iDPC–STEM maps. The atomic models were rigid-body fitted in the map, displayed and Figures prepared using UCSF Chimera[74], UCSF ChimeraX[70] and Coot[75].

**iDPC–STEM imaging simulations.** All STEM image simulations (applied in Fig. 1d,e and in Supplementary Fig. 4) were produced using the multi-slice method[52], extended to support the iDPC–STEM as explained in the following refs. [7,8,40,50] on several applications. The parameters for simulations were chosen to accommodate conditions given in Supplementary Fig. 4. Noise was added at the quadrant detection level based on electron dose used in the experiments and varied to illustrate the effect on the image.

**Reporting summary.** Further information on research design is available in the Nature Research Reporting Summary linked to this article.

## Data availability

All data needed to evaluate the conclusions in the paper are presented in the paper and/or the Supplementary Materials. KLH iDPC–STEM (3.5 mrad) and TMV iDPC–STEM (4.0 mrad) micrographs are publicly available as EMPIAR-11034 and EMPIAR-11042 data sets, respectively. KLH iDPC–STEM map was deposited at the Electron Microscopy Data Bank (EMDB) (EMD-14407). The EMDB accession numbers for the reconstructed TMV cryo-EM maps including fitted PDB coordinates are EMD-13778/PDB 7Q22 (CSA 2.0), EMD-13779/PDB 7Q23 (CSA 3.0), EMD-13780/PDB 7Q2A (CSA 3.5), EMD-13781/PDB 7Q2R (CSA 4.0) and EMD-13782/PDB 7Q2S (CSA 4.5).

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

## Acknowledgements

We thank the following people for their support during various stages: T. Hoffmann, A. Meingast, E. Yücelen, O.F. Raschdorf, S. Lazar, Y. Deng, R. Jonkers, B. Groen, B. de Vries, S. Vespucci and M. Veenis. We gratefully acknowledge the computing time granted through Jülich Aachen Research Alliance (JARA) on the supercomputer JURECA at Forschungszentrum Jülich[76]. K.M.-C. and M.-L.L. acknowledge funding from the Helmholtz Association under contracts VH-NG-1317 (moreSTEM) and ZT-I-0025 (Ptychography 4.0).

## Author contributions

I.L. initiated cryo-iDPC–STEM concept. I.L. and C.S. designed the research. M.W. optimized the STEM column alignment, adjusted imaging conditions for low-dose imaging and designed the acquisition workflow. M.W. and I.L. operated electron microscope in iDPC–STEM mode under these conditions. F.d.H. prepared TMV cryo-specimens and designed acquisition workflow. E.V.P. prepared KLH cryo-specimens. Initial data evaluation was performed by I.L., E.G.T.B., R.E., F.d.H. and E.V.P. Supporting simulations and related analysis were performed by E.G.T.B. Single-particle reconstruction of KLH was performed by D.M. and C.S. Image processing and TMV single-particle helical reconstruction was done by M.B., M.L.L., K.M.-C. and C.S. The paper, including figure preparation, was written by I.L., M.L.L., D.M and C.S. with input from all the authors.

## Funding

## Competing interests

I.L., M.W., F.d.H., E.V.P., R.E. and E.G.T.B. are employees of Thermo Fisher Scientific. The other authors declare no competing interests. The funders had no role in study design, data collection and analysis, decision to publish or preparation of the manuscript. The authors received no specific funding for this work.

## Additional information

**Correspondence and requests for materials** should be addressed to Ivan Lazić or Carsten Sachse.

# nature research

# Reporting Summary

Nature Research wishes to improve the reproducibility of the work that we publish. This form provides structure for consistency and transparency in reporting. For further information on Nature Research policies, see our Editorial Policies and the Editorial Policy Checklist.

## Statistics

For all statistical analyses, confirm that the following items are present in the figure legend, table legend, main text, or Methods section.

| n/a | Confirmed | |
|---|---|---|
| ☐ | ☒ | The exact sample size (*n*) for each experimental group/condition, given as a discrete number and unit of measurement |
| ☐ | ☒ | A statement on whether measurements were taken from distinct samples or whether the same sample was measured repeatedly |
| ☒ | ☐ | The statistical test(s) used AND whether they are one- or two-sided<br>*Only common tests should be described solely by name; describe more complex techniques in the Methods section.* |
| ☒ | ☐ | A description of all covariates tested |
| ☒ | ☐ | A description of any assumptions or corrections, such as tests of normality and adjustment for multiple comparisons |
| ☐ | ☒ | A full description of the statistical parameters including central tendency (e.g. means) or other basic estimates (e.g. regression coefficient) AND variation (e.g. standard deviation) or associated estimates of uncertainty (e.g. confidence intervals) |
| ☒ | ☐ | For null hypothesis testing, the test statistic (e.g. *F*, *t*, *r*) with confidence intervals, effect sizes, degrees of freedom and *P* value noted<br>*Give P values as exact values whenever suitable.* |
| ☒ | ☐ | For Bayesian analysis, information on the choice of priors and Markov chain Monte Carlo settings |
| ☒ | ☐ | For hierarchical and complex designs, identification of the appropriate level for tests and full reporting of outcomes |
| ☒ | ☐ | Estimates of effect sizes (e.g. Cohen's *d*, Pearson's *r*), indicating how they were calculated |

*Our web collection on statistics for biologists contains articles on many of the points above.*

## Software and code

Policy information about availability of computer code

| Data collection | Velox (ThermoFisher)v3.2, MAPS (ThermoFisher)v3.16 |
|---|---|
| Data analysis | EMAN2 2.91, SPRING 0.86.1661, RELION 3.1, CryoSPARC 3.2, UCSF Chimera 1.15, UCSF Chimera X 1.2, Coot |

For manuscripts utilizing custom algorithms or software that are central to the research but not yet described in published literature, software must be made available to editors and reviewers. We strongly encourage code deposition in a community repository (e.g. GitHub). See the Nature Research guidelines for submitting code & software for further information.

## Data

Policy information about availability of data

All manuscripts must include a data availability statement. This statement should provide the following information, where applicable:
- Accession codes, unique identifiers, or web links for publicly available datasets
- A list of figures that have associated raw data
- A description of any restrictions on data availability

The following publicly available data were used in the manuscript: for TMV, 2015/2019 CTEM data sets (EMPIAR-10021 and EMPIAR-10305) and PDB coordinates of (PDB-ID 4UDV and 6SAG); for KLH, we docked the available structure PDB-ID 4BED into the iDPC-STEM map.

All data needed to evaluate the conclusions in the paper are presented in the paper and/or the Supplementary Materials. KLH iDPC-STEM (3.5 mrad) and TMV iDPC-STEM (4.0 mrad) are available as EMPIAR-11034 and EMPIAR-11042 data sets, respectively. KLH iDPC-STEM map was deposited at the EMDB (EMD-14407). The EMDB accession numbers for the reconstructed TMV cryo-EM maps including fitted PDB coordinates are EMD-13778/PDB-ID 7Q22 (CSA:2.0), EMD-13779/PDB-ID 7Q23 (CSA: 3.0), EMD-13780/PDB-ID 7Q2A (CSA: 3.5), EMD-13781/PDB-ID 7Q2R (CSA: 4.0) and EMD-13782/PDB-ID 7Q2S (CSA: 4.5).

# Field-specific reporting

Please select the one below that is the best fit for your research. If you are not sure, read the appropriate sections before making your selection.

☒ Life sciences ☐ Behavioural & social sciences ☐ Ecological, evolutionary & environmental sciences

For a reference copy of the document with all sections, see nature.com/documents/nr-reporting-summary-flat.pdf

# Life sciences study design

All studies must disclose on these points even when the disclosure is negative.

| | |
|---|---|
| Sample size | Two different biological specimens were imaged, KLH and TMV. For KLH, two data sets of 760 and 687 micrographs (2.0 and 3.5 mrad) and for TMV, a total of 5 different data sets were acquired at convergence semi-angles (CSA) of 2.0, 3.0, 3.5, 4.0 and 4.5 with a total of 20, 13, 15, 20 and 28 micrographs. The number of micrographs was determined by the number of extractable particles for KLH or segments for TMV for further image processing. |
| Data exclusions | Micrographs of poor particle coverage and ice quality were discarded. |
| Replication | Due to the time-consuming nature of image acquisition and the limited access to this specialized microscope equipment (Krios and STEM unit), exact replicates were not performed. |
| Randomization | Randomization was not applicable in the current study because of the time-consuming nature of image acquisition and the limited access to this specialized microscope equipment (Krios and STEM unit). |
| Blinding | Blinding experiments were not applicable to the current study because of the time-consuming nature of image acquisition and the limited access to this specialized microscope equipment (Krios and STEM unit). |

# Reporting for specific materials, systems and methods

We require information from authors about some types of materials, experimental systems and methods used in many studies. Here, indicate whether each material, system or method listed is relevant to your study. If you are not sure if a list item applies to your research, read the appropriate section before selecting a response.

## Materials & experimental systems

| n/a | Involved in the study |
|---|---|
| ☒ ☐ | Antibodies |
| ☒ ☐ | Eukaryotic cell lines |
| ☒ ☐ | Palaeontology and archaeology |
| ☒ ☐ | Animals and other organisms |
| ☒ ☐ | Human research participants |
| ☒ ☐ | Clinical data |
| ☒ ☐ | Dual use research of concern |

## Methods

| n/a | Involved in the study |
|---|---|
| ☒ ☐ | ChIP-seq |
| ☒ ☐ | Flow cytometry |
| ☒ ☐ | MRI-based neuroimaging |

