## [Peer Review File · Nature Methods]

Peer Review Information

Manuscript Title: Single-particle cryo-EM structures from iDPC-STEM at near-atomic resolution

Corresponding author name(s): Carsten Sachse

Reviewer Comments & Decisions:

Decision Letter, initial version:
--

Dear Professor Sachse,

Thank you for submitting your manuscript entitled "Single-particle cryo-EM structures from iDPC-STEM at near-atomic resolution". We have given the paper our careful consideration but we regret that we cannot publish it in Nature Methods in its current form.

Among the considerations that arise at this stage are a manuscript's probable interest and immediate practical relevance to a general readership. We do not doubt the technical quality of your work or that it will be of interest to others working in this area of structural biology research. However, we do not think that the technical advances presented in the current version of the manuscript will have a sufficiently significant impact on a broader readership to justify publication in Nature Methods.

Should future experimental data allow you to add a additional demonstration, ideally to solve a protein structure, we would be happy to look at a revised manuscript (unless, of course, something similar has by then been accepted at Nature Methods or appeared elsewhere). This includes submission or publication of a portion of this work somewhere else. In the case of eventual publication, the received date would be that of the revised paper.

If you are interested in submitting a suitably revised manuscript in the future or if you have any questions, please contact me.

Thank you for your interest in Nature Methods. I am sorry that on this occasion we cannot be more positive.

Sincerely,
Arunima

Arunima Singh, Ph.D.

Senior Editor
Nature Methods

Author Rebuttal to Initial comments

Dear Arunima,

Thank you for your patience regarding the manuscript inquiry on a second potential application of the iDPC-STEM method to cryo-EM structure determination.

At your request, we now rapidly assessed an additional iDPC-STEM data set on the keyhole limpet hemocyanin (KLH) protein, for which an initial micrograph was also presented in Figure 1h and 1i of the submitted manuscript. The presented KLH results should be considered work in progress that has not reached the same level

of maturity of the TMV results, yet. We processed only a total of 54 micrographs and obtained a 11.0 Å resolution map from only 5,230 particles (**Figure A1**). The map agrees well with the expected map features and the fitted PDB coordinates. Knowing that the currently highest resolution structure of KLH was published at 9 Å

(Gatsogiannis and Marki, *J Mol Biol* 2009, EMD-1569)- this is an encouraging result. Nevertheless, in this short term we did not have time to optimize the acquisition and perform the analysis in equal depth as presented for TMV. We believe that the small

amount of recordable data is also a limiting factor. At the same, as the currently highest resolution map is only 9 Å, there may be additional sample-limiting factors that prevent any further improvement of the structure. This is another reason why

we preferred TMV as a test specimen as it was shown to be resolvable down to 1.9 Å using cryo-EM (Weis et al., *EMBO Rep* 2019).

Figure A: Summary of keyhole limpet hemocyanin (KLH) iDPC-STEM data set. Top left. Typical micrographs recorded at 2.7 mrad. Top right. Five class averages of KLH particles. iDPC-STEM cryo-EM map at 11.0 Å resolution with corresponding Fourier Shell Correlation. Bottom. Available POB coordinates (POB ID 4BEO) could be docked into the cryo-EM density (left. side and top view, right. segmented density of different subunits).

Given these uncertainties, we feel that the addition of a second data set to the manuscript is currently not preferable. We believe that with the proper time spent on acquisition, we will obtain a KLH map with better resolution than currently published. In the current manuscript, we presented a strong case for the overall feasibility and the first near-atomic resolution map obtained by iDPC-STEM using TMV as a test specimen. We wonder whether your requested criterion is really commonplace and should be applied to these cumbersome structural biology methods. In order to explain my perception in more detail, I will point out three comparable examples of past high-profile cryo-EM method manuscripts that showcase the application to a "single protein structure" rather than two requested different ones while the manuscripts still generated a big impact:

1. Yip, et al. *Nature*, vol. 587, no. 7832, pp. 157-161, Nov. 2020. investigate the use of novel imaging hardware for cryo-EM using a single apoferritin protein structure and report a very high-resolution structure.
2. Tegunov, et al." *Nat Methods*, vol. 18, no. 2, pp. 186-193, Feb. 2021. demonstrated the utility of using subtomogram averaging to obtain near-atomic resolution structures of ribosomes in the cellular environment. Again, a single protein structure is resolved.
3. The use of STEM imaging to cryo-frozen samples in biology was put forward in a *Nature Methods* publication of Wolf, et al. *Nat Methods*, vol. 11, no. 4, pp. 423-428, Apr. 2014. Again, the imaging of a single cell line was demonstrated, not several different ones.

In our submitted manuscript, we systematically imaged five different samples of the TMV specimen under different imaging conditions (5 rows of Figure 4). Five different samples and five different data sets independently showcase the principal performance and utility of the iDPC-STEM approach to investigate and determine biological structures. We chose TMV as a specimen as TMV has a long history in structural biology as a whole (used since the 1930s) and in cryo-EM in particular all of which demonstrate that results obtained with TMV are very good general indicator for the performance of the technique (Table 3 of the submitted manuscript).

We took your request very seriously and openly discussed with you our ongoing research. We believe that your formal requirement is achievable but not within this short time. At the same time, we presented a sound case for the future potential of the iDPC-STEM technique using TMV. We believe it is worth to be considered for publication at *Nature Methods*, as such a resolution was never achieved by any STEM technique on any biological sample before. Given these considerations, please, let us know how you would like to proceed with the submitted manuscript.

Sincerely yours,

Prof. Dr. Carsten
Sachse On behalf of
all co-authors

Decision Letter, first revision:

Dear Professor Sachse,

Your Article entitled "Single-particle cryo-EM structures from iDPC-STEM at near-atomic resolution" has now been seen by 3 reviewers, whose comments are attached. While they find your work of potential interest, they have raised serious concerns which in our view are sufficiently important that they preclude publication of the work in Nature Methods, at least in its present form.

As you will see, the reviewers raise concerns about the paper lacking proper comparisons to conventional approaches, a clear demonstration of practical advantage and generalizability of the approach as well as more detailed description of the method such that it can be adapted by other users.

We and the reviewers agree that this is a very neat demonstration, but as discussed previously, we think you need to make a strong case for the advance the method offers for biological research, which will require a more challenging demonstration. Should further experimental data allow you to fully address these criticisms we would be willing to look at a revised manuscript (unless, of course, something similar has by then been accepted at Nature Methods or appeared elsewhere). This includes submission or publication of a portion of this work somewhere else. We hope you understand that until we have read the revised paper in its entirety we cannot promise that it will be sent back for peer-review.

Alternatively, if you like, I would be happy to consult with my colleagues at Nature Communications about the possibility of a transfer of a suitably revised manuscript to the journal.

If you are interested in revising this manuscript for submission to Nature Methods in the future, **please contact me to discuss your appeal and revision plan before making any revisions.** Otherwise, we hope that you find the reviewers' comments helpful when preparing your paper for submission elsewhere.

Sincerely,
Arunima

Arunima Singh, Ph.D.
Senior Editor

Nature Methods

Reviewers' Comments:

Reviewer #1:

Remarks to the Author:

Lazic et al. report an iDPC-STEM application to frozen hydrated TMV samples. A 300kV Titan Krios G4 equipped with a Panther segmented STEM detector was used and STEM datasets over 500 nm (or 700 nm?) scan range with varying convergence semi angles were recorded of frozen virus samples. Image processing with the Relion3.1 software resulted in 3D reconstructions at different resolutions as a function of the convergence semi angles of the illumination. Under the best illumination conditions in this work of 4 mrad, a final map at 3.5Å resolution was obtained, which is comparable to the resolution achieved by Fromm et al. (2015) from the same Sachse group, when using a Falcon II direct electron detector in real-space conventional TEM imaging mode.

This work impressively documents that iDPC-STEM has to be considered as an alternative to CTEM data collection for frozen hydrated specimens. As such, this is an important milestone paper. The main impact of this work will be with the community that applies Conventional TEM (CTEM) to frozen hydrated samples. This community is not very familiar with STEM. As this manuscript is the first significant contribution to the structural analysis of cryo-EM samples using STEM, the manuscript should be written so that the CTEM community can understand it.

It is not clear why the iDPC-STEM data were not compared with K2 data of Weis et al. (2019), which reached higher resolution. iDPC-STEM from today should be compared to CTEM from today, not from 6 years ago.

The quantitative comparison of the strengths and weaknesses of iDPC-STEM could be done more rigorously.

iDPC-STEM scans the electron beam in a 700nm long line over the sample, before returning to collect the next scan line. (Does the beam go back-and-forth over the sample, or forth-and-forth, i.e., the return is without significant beam exposure?)

In STEM mode, any protein of a few nm dimensions will therefore be exposed by the electron beam in several scan lines that are separated by time. Structural STEM data are collected of a protein "after" the adjacent part of the protein has already been fully beam-damaged by 35 e/Å². In contrast, in CTEM mode, data are collected "while" beam damage occurs, and importantly the first frames of a dose-fractionated movie-mode real-space imaging collects data on the entire protein that is still in a non-beam damaged state. This in the viscous environment of vitrified water may or may not make a difference to the SNR vs. electron beam damage of recorded data. This manuscript could discuss this fundamental difference between the two illumination modes.

In CTEM, the accumulating beam damage can be quantified in B-factor curves for each movie frame. In this manuscript, the authors mention the possibility of dose-fractionated iDPC-STEM, by which the authors presumably mean to perform several STEM scans of the same sample region (line 305). The authors could discuss the time requirements for such multiple-scan STEM approach.

In CTEM, sample drift over time or over electron dose affects collected data in form of particle movements, which are tracked and corrected by image processing of dose-fractionated movies.

In STEM, sample drift over time results in image distortions such as skewing or anisotropic magnification, and sample drift over electron dose will result in resolution loss of projected particle images. The authors report that their STEM data do not appear to suffer critically from beam-induced movement. How do they quantify this? Do datasets recorded at different electron fluences show different features in terms of Guinier B-factors or particle distortions or resolution losses? This could also be discussed.

In CTEM, the performance of the entire imaging chain (consisting of electron source, lenses, sample identity and quality, and detector) can be quantified by the ResLog B-factor calculated from final data sets. Conventionally, instrument performances in CTEM are compared with each other when using apoferritin as test sample. This ResLog B-factor reports on the SNR of recorded particles. This is different than the Guinier B-factor that reports on the amplitude decay towards higher resolution. Ideally, the iDPC-STEM method should be compared to CTEM by establishing a measure of the ResLog B-factor of this method when applied to apoferritin. If the authors are not able to perform that in the scope of this manuscript, then they should at least calculate a ResLog B-factor for TMV from a comparable instrument (with a current dataset, not one from 2015) and compare that with a ResLog B-factor for these iDPC-STEM data.

In CTEM, contrast is maximized by optimizing the defocus setting, whereby the operator attempts to balance the phase contrast vs. resolution of the instrument's CTF.

In iDPC-STEM as applied here, the tool records a significant amount of phase contrast. The CTF of iDPC-STEM thereby sensitively depends on the defocus setting. With a 50nm thick sample protein sample, a particle at the top and at the bottom surface of the ice layer should have a significantly different CTF, which is later not fitted in Relion and which therefore should limit resolution. In contrast to CTEM, where defocus is later fitted during image processing and where the defocus difference between top and bottom parts of the same particle are corrected through Ewald sphere curvature correction, accurate focusing of the iDPC-STEM during data collection is important for reaching high resolution. This should be discussed here.

How does the phase contrast CTF of a CTEM compare with the phase contrast CTF of iDPC-STEM? It is clear that iDPC-STEM CTF curves are sample dependent and dependent on detector dimensions and defocus, so that an objective comparison of the two CTF curves is difficult to provide. However, as this is the main message of this manuscript, the CTF performance of the two should at least be discussed quantitatively, including a discussion of the SNR vs. beam damage at higher resolution between the two methods.

According to TFS documentation, the Panther STEM detector consists of two different detectors, one with eight quadrants surrounding a central hole, and one with eight quadrants without hole.

What was the scan direction with respect to the orientation of the four-quadrant detector?

This manuscript suggests that the detector with the central hole was used here. Is that so? On the other side, you write in line 224 that iDPC-STEM uses all the available electrons, suggesting that you used the detector without the hole. Which one is it?

If the one with the hole was used: At the different illumination convergence semi angles used, what were the inner and outer radii in mrad of the inner and outer quadrant segments of the Panther STEM detector used here? Which quadrants were used, the four inner or the four outer segments, or the four combinations of inner and outer segments? How was the signal from the quadrants evaluated to produce the signal, i.e., how does the system calculate the center of mass movement? How sensitive is this when single electron counting with very few electrons is applied?

Reviewer #2:

Remarks to the Author:

The manuscript by Lazic et al demonstrates that iDPC-STEM can be a useful tool in cryo-EM to determine the structure of biomolecules at high resolution. Quoting from the paper, "This study shows that iDPC-STEM imaging can be successfully applied to cryo-EM single-particle based structure determination and elucidate biological structures at near-atomic resolution." There is a long-standing notion in life science EM that STEM is not useful for cryo-imaging, while the materials scientists have been charging ahead with cryo-STEM and have reached spectacular resolutions on sensitive materials with very low dose. Until now cryo-STEM in life science was applied mainly to tomography of thick samples, where cryo-TEM fails due to inelastic scattering. Expansion into high resolution on molecular specimens depends on phase contrast, which is implemented in the iDPC technique developed by Lazic and co-workers at the FEI/TFS research labs. According to the contributions list he collected the data himself at the lab of Sachse in Juelich that is very experienced in single particle analysis and especially in TMV studies. This seems a good recipe for success. iDPC has many known advantages that are exploited in materials science, even if it is not yet entirely routine practice. Known advantages include the visibility of heavy elements next to light ones (unlike HAADF) and the availability of sub-atomic resolution at very low dose. Complete contrast transfer is also known, at least theoretically, because the images are acquired in focus. The current manuscript demonstrates that very impressively. The absence of beam-induced sample movement is a new result as far as I'm aware. That is very encouraging for the further development of STEM methods for cryo-EM. On technical grounds the work is a major step forward.

As a Nature Methods paper I'm ambivalent. It's not clear from the work what advantages iDPC-STEM holds over the current state of the art. Why would one choose that method instead of existing TEM in order to get the best data? The CTF contrast inversions in TEM are annoying, but they are corrected very successfully by standard image processing pipelines. Is the iDPC method potentially faster, or cheaper, or easier than conventional TEM? Can it be applied more widely? In fact TMV is a very particular specimen. All the particles lie flat and almost perfectly ordered along the grid. There is no concern about preferential orientations or slight differences in defocus per particle. In this way TMV is a relatively easy case. The manuscript shows very nicely that iDPC-STEM produces high resolution images that are amenable to single particle processing for TMV. It does not show that iDPC-STEM would be a method of choice for someone who already has a Krios-level microscope and DED camera, nor would it convince someone interested in molecular structure "as fast as possible" to invest in iDPC rather than the best TEM camera. The manuscript leaves me wondering, what can it do that TEM cannot? The conclusions are sound, but more scientific than practical. That stated, publication in Nature Methods is a decision for the editors. More technical comments follow.

1. The iDPC detector is not standard equipment, at least not in life science EM. It deserves a more complete description than the single sentence on page 12. A drawing would be useful, or at least a link to a full description. The heart of the new method is left as a commercial black box.

2. Fig 2 is very impressive. Please clarify if the images were processed in any way, for example background correction or high pass filtering. The texture differs from the panels of Fig 3 on the left,

and looks more like the filtered images under the heading Segmented Particles on the right.

3. The data were acquired on a Krios microscope that has a Falcon 4 DED camera installed. It was used for screening. A side by side comparison of the iDPC-STEM with the Falcon images at high magnification would be useful and convincing. It would also be interesting to see a comparison with conventional HAADF images (if those are already available).

4. It is strange that the TEM benchmark was the dataset EMPIAR-10021 from 2015 in Ref 40, rather than the more recent study of the same group in 2019 (Ref 33, EMPIAR-10305) at 1.9Å. This is the current state of the art. Most likely the data also come from the same microscope since the correspondence address is in Juelich rather than EMBL. The only comment in the text is "As a defocused-based CTEM reference data set, we used a previously determined TMV structure, EMPIAR-10021 [40]."

5. Analysis of the resolution and contribution of high spatial frequencies using layer line intensity ratios as a function of CSA is elegant and informative. It's also interesting that the STEM resolution in the reconstruction came out slightly better than the TEM for a comparable number of particles, implying that the information content per particle is higher in iDPC-STEM. This could be pursued a bit further by comparing reconstructions from subsets of the data. (an entirely optional suggestion)

6. Depth sectioning with aberration correction should include reference to deJonge, for example <https://doi.org/10.1017/s1431927611012347>.

7. The paragraph in the discussion on aliasing (highlighted in yellow) describes a new result. It should better go into the results section, or maybe as an explanatory note in Supplementary Material. The bottom line is that the pixel spacing should fully sample the optical resolution, which is no surprise really. The semantics of cutoffs smaller or larger than Nyquist get confusing, and this is a distraction in the important discussion section.

Reviewer #3:

Remarks to the Author:

Lazic and colleagues report the first high-resolution results from the application of iDPC-STEM (and STEM in general) to cryogenically frozen biological samples. This is an outstanding achievement and will be of great interest to researchers and manufacturers in the electron microscopy field. However, at this stage the technique does not appear to offer practical and theoretical advantages over the conventional cryo-EM approach, and therefore will be of limited interest to the broader structural biology community. Several important aspects that could have made the work more attractive should have been discussed in detail:

1. In terms of performance, does iDPC-STEM have the potential to provide benefits over bright field cryo-EM? These may include higher resolution, higher throughput, access to more difficult samples, additional layers of quantitative information etc.

2. In practical terms, what are the perspectives of iDPC-STEM for automation, ease of use, cost-effective and smaller footprint instrumentation etc.?

3. Overall, the authors should give an outlook on the future of the method and its potential advantages.

Author Rebuttal, first revision:

Dear Arunima,

Thank you for your assessment on our Nature Methods manuscript. After discussion with the co-authors, we feel that we are indeed in a position to properly address the raised points and generate a revised manuscript. Therefore, I attach a preliminary response to the referees where we outline our planned experiments and responses. In this context, we would appreciate any intermediate feedback in case we might have misinterpreted the referee's and your points.

Many thanks for your feedback on the matter.

Best wishes,

Carsten

Preliminary Response and planned experiments (NMETH-A46924A-Z)

We appreciate the overall positive feedback on our cryo-iDPC-STEM manuscript. The three reviewers asked well-deserved questions in an effort to widen the scope of the manuscript, which we fully endorse. In this preliminary response, we outline how we plan to address the raised points.

The proposed experiments and analyses are designed in such a way that they can be completed with a three-months revision time frame. We feel that we can address most of the points within this time, while we also point out the ones that go beyond the scope of the manuscript. In summary, we want to subject another specimen to our structure determination pipeline in order to demonstrate the wider general applicability of the cryo-iDPC-STEM approach. Other requests include additional data processing with an updated reference data set of TMV for better comparison, improved result presentation and extended discussions on the principal benefits of the imaging approach.

Please find our point-by-point response below.

Reviewer #1:

Remarks to the Author:

This work impressively documents that iDPC-STEM has to be considered as an alternative to CTEM data collection for frozen hydrated specimens. As such, this is an important milestone paper.

1. The main impact of this work will be with the community that applies Conventional TEM (CTEM) to frozen hydrated samples. This community is not very familiar with STEM. As this manuscript is the first

significant contribution to the structural analysis of cryo-EM samples using STEM, the manuscript should be written so that the CTEM community can understand it.

We agree with the Reviewer that in the revised manuscript we will make an extra-effort to describe the technical details of the STEM data acquisition in more detail for further clarification. Therefore, we plan to include an additional paragraph on data acquisition parameters in the Results section.

2. It is not clear why the iDPC-STEM data were not compared with K2 data of Weis et al. (2019), which reached higher resolution. iDPC-STEM from today should be compared to CTEM from today, not from 6 years ago.

We intentionally chose the older data set due to better resolution comparability as the resolution is very close to the one obtained by the iDPC-STEM method. We fully agree, however, with the Reviewer's remark (see below Reviewer 2, point 5) that we should also compare it with our latest structure detailed by Weis et al. when we compare CTEM and STEM in general. Therefore, we will include the data set and structure in the comparisons of Figure 3 and Figure 4.

The quantitative comparison of the strengths and weaknesses of iDPC-STEM could be done more rigorously.

3. iDPC-STEM scans the electron beam in a 700nm long line over the sample, before returning to collect the next scan line. (Does the beam go back-and-forth over the sample, or forth-and- forth, i.e., the return is without significant beam exposure?)

As requested, we will detail the exact geometry sequence of the scan in a new paragraph of the Results section (Reviewer 1, point 1). The below addressed points 10, 11 and 12 on the detector will also be helpful for the better understanding of this matter.

4. In STEM mode, any protein of a few nm dimensions will therefore be exposed by the electron beam in several scan lines that are separated by time. Structural STEM data are collected of a protein "after" the adjacent part of the protein has already been fully beam- damaged by 35 e/A². In contrast, in CTEM mode, data are collected "while" beam damage occurs, and importantly the first frames of a dose-fractionated movie-mode real-space imaging collects data on the entire protein that is still in a non-beam damaged state. This in the viscous environment of vitrified water may or may not make a difference to the SNR vs. electron beam damage of recorded data. This manuscript could discuss this fundamental difference between the two illumination modes.

As suggested by the Reviewer, we will raise this critical difference between the imaging strategies of CTEM vs STEM in the Discussion section of the revised manuscript. Indeed, the scanning mode causes different mechanisms of energy dissipation and, therefore, offers opportunities for different electron delivery strategies when compared with typical parallel illumination approaches.

5. In CTEM, the accumulating beam damage can be quantified in B-factor curves for each movie frame. In this manuscript, the authors mention the possibility of dose-fractionated iDPC- STEM, by which the authors presumably mean to perform several STEM scans of the same sample region (line 305). The

authors could discuss the time requirements for such multiple- scan STEM approach.

As requested by the Reviewer, we plan to include the differences between the typical movie mode of CTEM acquisition vs. a dose-fractionated mode implemented in STEM in the Discussion.

6. In CTEM, sample drift over time or over electron dose affects collected data in form of particle movements, which are tracked and corrected by image processing of dose- fractionated movies. In STEM, sample drift over time results in image distortions such as skewing or anisotropic magnification, and sample drift over electron dose will result in resolution loss of projected particle images. The authors report that their STEM data do not appear to suffer critically from beam-induced movement. How do they quantify this? Do datasets recorded at different electron fluences show different features in terms of Guinier B-factors or particle distortions or resolution losses? This could also be discussed.

We agree with the Reviewer that more care should be taken to delineate the different effects of beam-induced motion in CTEM vs. STEM imaging. Based on the imaging geometry, we plan to analyze different quadrants of the micrographs corresponding to different exposure time points and correlate them with imaging quality parameters. Depending on the outcome of the analysis, we will add the analysis to the Results if appropriate. In any case, we will add this item to the Discussion as it represents a fundamental difference in the way beam-induced motion occurs in the different imaging modes. It may provide a direct explanation why STEM without dedicated motion correction was able to achieve the same resolution as CTEM with motion correction.

7. In CTEM, the performance of the entire imaging chain (consisting of electron source, lenses, sample identity and quality, and detector) can be quantified by the ResLog B-factor calculated from final data sets. Conventionally, instrument performances in CTEM are compared with each other when using apoferritin as test sample. This ResLog B-factor reports on the SNR of recorded particles. This is different than the Guinier B-factor that reports on the amplitude decay towards higher resolution.

Ideally, the iDPC-STEM method should be compared to CTEM by establishing a measure of the ResLog B-factor of this method when applied to apoferritin. If the authors are not able to perform that in the scope of this manuscript, then they should at least calculate a ResLog B- factor for TMV from a comparable instrument (with a current dataset, not one from 2015) and compare that with a ResLog B-factor for these iDPC-STEM data.

As requested by the Reviewer, we performed the B-factor calculation based on the Rosenthal- Henderson ResLog plot with our best resolution STEM 4.0 mrad and CTEM data set (Figure attached below). The obtained B-factors confirm that iDPC-STEM at 4.0 mrad shows a better performance than CTEM (EMPIAR-10021) from 6 years ago. We will update the analysis of the latest TMV data set (EMPIAR-10305).

8. In CTEM, contrast is maximized by optimizing the defocus setting, whereby the operator attempts to balance the phase contrast vs. resolution of the instrument's CTF.

In iDPC-STEM as applied here, the tool records a significant amount of phase contrast. The CTF of iDPC-STEM thereby sensitively depends on the defocus setting. With a 50nm thick sample protein sample, a particle at the top and at the bottom surface of the ice layer should have a significantly different CTF, which is later not fitted in Relion and which therefore should limit resolution. In contrast to CTEM, where defocus is later fitted during image processing and where the defocus difference between top and bottom parts of the same particle are corrected through Ewald sphere curvature correction, accurate focusing of the iDPC-STEM during data collection is important for reaching high resolution. This should be discussed here.

We thank the Reviewer for bringing up this topic as it arises from a misunderstanding. In the thickness-of-the-sample treatment of the Discussion (page 8), the key parameter to be considered is the depth of focus of the beam. In the CSAs used throughout the manuscript, the depth of focus is significantly larger than the sample thickness, therefore CTF variations within the sample do not need to be considered. This problem will become more critical when larger CSAs such as 10 mrad are used as the resulting depth of focus is 39 nm may become smaller than the ice thickness. We will add a clarifying statement to the relevant manuscript section.

9. How does the phase contrast CTF of a CTEM compare with the phase contrast CTF of iDPC-STEM? It is clear that iDPC-STEM CTF curves are sample dependent and dependent on detector dimensions and defocus, so that an objective comparison of the two CTF curves is difficult to provide. However, as this is the main message of this manuscript, the CTF performance of the two should at least be discussed quantitatively, including a discussion of the SNR vs. beam damage at higher resolution between the two methods.

As requested by the Reviewer, we will add a comparative graph on the typical CTEM-CTF including phase contrast employed in the reference TMV data sets in addition to the existing STEM-CTF Figure S2e. Following this analysis, we will also take up the SNR considerations at high resolution in the Discussion to further delineate the differences between STEM and CTEM. Briefly, for thin samples iDPC-STEM CTF are not sample dependent, while detector dimensions and defocus dependencies are well understood, analytically expressed, and explained in full depth [Lazic, Bosch book chapter 2017]. Thick samples effects are also studied in depth and reported [Bosch, Lazic 2019].

10. According to TFS documentation, the Panther STEM detector consists of two different detectors, one with eight quadrants surrounding a central hole, and one with eight quadrants without hole.

We had the Panther 4-quadrant detector installed as specified in the Results section (page 4) of the manuscript. We will highlight this point once more in the dedicated Results section (Reviewer 1, point 1). The 8-segment detector arm is used in the basic 4-quadrant mode (inner to outer radius ratio 1/6), where quadrant segments are hard wired (no use is made of the radial separation) (Figure below). By adjusting camera length, the central hole radius is always set to be at least 4 times smaller than the bright field radius (beam size at the detector). The second type of detector (no hole detector arm) is not enabled in the Panther version we used.

11. What was the scan direction with respect to the orientation of the four-quadrant detector?

The new Results section will clarify the scanning geometry (Reviewer 1, point 1). Briefly, the initial offset angle between the scanning coordinate frame and the detector coordinate frame is measured and set (angle θ). Once the angle is set during the alignment, the scan direction can be chosen freely. The rotation angle will be recorded and the coordinate frame will be adjusted automatically (Figure below). This is a straightforward operation, yet it is critical to perform final integration step in iDPC-STEM correctly.

12. This manuscript suggests that the detector with the central hole was used here. Is that so? On the other side, you write in line 224 that iDPC-STEM uses all the available electrons, suggesting that you used the detector without the hole. Which one is it?

If the one with the hole was used: At the different illumination convergence semi angles used, what were the inner and outer radii in mrad of the inner and outer quadrant segments of the Panther STEM detector used here? Which quadrants were used, the four inner or the four outer segments, or the four combinations of inner and outer segments? How was the signal from the quadrants evaluated to produce the signal, i.e., how does the system calculate the center of mass movement? How sensitive is this when single electron counting with very few electrons is applied?

In order to address this question, we will clarify this point in the detailed detector description. The effect of the hole is well studied, understood and documented. As mentioned above, the detector is used in the 4-quadrant mode, where segments are hard wired and no use is made of the radial separation of Panther. Indeed, the central hole is present while ideally it should not be. Nevertheless, when small, the hole has practically no effect. By adjusting camera length, the central hole radius is always set to be at least 4 times smaller than the bright field radius (beam size at the detector). The effect of the hole on the CTF is well understood and explained in [Lazic, Bosch book chapter 2017]. The central part of the CBED pattern changes minimally while scanning over atomic columns [Lazic, Bosch Lazar 2016] and therefore does not contribute to the CBED pattern center of mass movement significantly. Up to 30% radius of the BF disk, the hole has a negligible effect. Therefore, even when some electrons are lost there, they would contribute mostly to the noise. Previous results published in literature so far have been generated with detector including middle hole.

Reviewer #2:

Remarks to the Author:

The manuscript by Lazic et al demonstrates that iDPC-STEM can be a useful tool in cryo-EM to determine

the structure of biomolecules at high resolution. Quoting from the paper, "This study shows that iDPC-STEM imaging can be successfully applied to cryo-EM single-particle based structure determination and elucidate biological structures at near-atomic resolution."

There is a long-standing notion in life science EM that STEM is not useful for cryo-imaging, while the materials scientists have been charging ahead with cryo-STEM and have reached spectacular resolutions on sensitive materials with very low dose. Until now cryo-STEM in life science was applied mainly to tomography of thick samples, where cryo-TEM fails due to inelastic scattering. Expansion into high resolution on molecular specimens depends on phase contrast, which is implemented in the iDPC technique developed by Lazic and co-workers at the FEI/TFS research labs. According to the contributions list he collected the data himself at the lab of Sachse in Juelich that is very experienced in single particle analysis and especially in TMV studies. This seems a good recipe for success. iDPC has many known advantages that are exploited in materials science, even if it is not yet entirely routine practice. Known advantages include the visibility of heavy elements next to light ones (unlike HAADF) and the availability of sub-atomic resolution at very low dose. Complete contrast transfer is also known, at least theoretically, because the images are acquired in focus. The current manuscript demonstrates that very impressively. The absence of beam-induced sample movement is a new result as far as I'm aware. That is very encouraging for the further development of STEM methods for cryo-EM. On technical grounds the work is a major step forward.

1. a) As a Nature Methods paper I'm ambivalent. It's not clear from the work what advantages iDPC-STEM holds over the current state of the art. Why would one choose that method instead of existing TEM in order to get the best data? The CTF contrast inversions in TEM are annoying, but they are corrected very successfully by standard image processing pipelines. Is the iDPC method potentially faster, or cheaper, or easier than conventional TEM?

b) It does not show that iDPC-STEM would be a method of choice for someone who already has a Krios-level microscope and DED camera, nor would it convince someone interested in molecular structure "as fast as possible" to invest in iDPC rather than the best TEM camera. The manuscript leaves me wondering, what can it do that TEM cannot? The conclusions are sound, but more scientific than practical. That stated, publication in Nature Methods is a decision for the editors. More technical comments follow.

The Reviewer acknowledges that the presented work represents a major technical step forward. At the same time, the Reviewer raises a number of highly interesting points regarding the practical consequences, e.g. whether iDPC-STEM is faster, cheaper and easier than CTEM or even the method of choice between the two. We feel that this point cannot be easily addressed within the scope of this manuscript as it would require an entirely new study including optimized hardware design with speed and ease of use in mind. We feel that we cannot compare the cryo-CTEM level of maturity and productivity including the current speed and ease of use at this point in time considering the amount of development that went into cryo-CTEM development. We believe, however, that this manuscript may mark the beginning of such a development as it demonstrates the principal applicability of the technique.

c) Can it be applied more widely? In fact TMV is a very particular specimen. All the particles lie flat and almost perfectly ordered along the grid. There is no concern about preferential orientations or slight differences in defocus per particle. In this way TMV is a relatively easy case. The manuscript shows very nicely that iDPC-STEM produces high resolution images that are amenable to single particle processing for TMV.

TMV is indeed a very good test specimen for cryo-EM imaging in particular and structural biology in general. In order to demonstrate that high resolutions of other specimens can be obtained and a wider range of specimens can be imaged, we plan to include a high-resolution data set on a second specimen to reach a similar resolution. Preliminary results of the KLH specimen are included in the Figure below. Additional data needs to be collected in order to improve the resolution further.

2. The iDPC detector is not standard equipment, at least not in life science EM. It deserves a more complete description than the single sentence on page 12. A drawing would be useful, or at least a link to a full description. The heart of the new method is left as a commercial black box.

As requested by the Reviewers, we will include the detector/scanning scheme drawing in the Methods section (Reviewer 1, points 10-12) and more detailed description on scanning and detection / acquisition process (Reviewer 1, point 1). As mentioned above, we will comment and clarify on detector geometry, scanning vs. detector orientation, effect of the hole to the CTF and related camera length choice.

3. Fig 2 is very impressive. Please clarify if the images were processed in any way, for example background correction or high pass filtering. The texture differs from the panels of Fig 3 on the left, and looks more like the filtered images under the heading Segmented Particles on the right.

We agree with the Reviewer that the exact treatment of the images has a significant impact on the appearance and visual interpretability of the micrographs. We will add a clarifying statement to the Figure legend and highlight this pre-processing routine in the Results section.

4. The data were acquired on a Krios microscope that has a Falcon 4 DED camera installed. It was used for screening. A side by side comparison of the iDPC-STEM with the Falcon images at high magnification would be useful and convincing. It would also be interesting to see a comparison with conventional HAADF images (if those are already available).

As requested, we plan to include a new STEM data set (Reviewer 2, point 1b) of a different specimen than TMV. In addition, we plan to present the same particle images taken by CTEM with a Falcon 4 DED camera. Simultaneous imaging in STEM is another advantage over CTEM that we will present in the revised manuscript. We will add an example of ADF-, (A)BF- and iDPC-STEM micrograph next to each other in the Supplementary section and point out the absence of high-resolution information in the first two (Figure below).

5. It is strange that the TEM benchmark was the dataset EMPIAR-10021 from 2015 in Ref 40, rather than

the more recent study of the same group in 2019 (Ref 33, EMPIAR-10305) at 1.9Å. This is the current state of the art. Most likely the data also come from the same microscope since the correspondence address is in Juelich rather than EMBL. The only comment in the text is "As a defocused-based CTEM reference data set, we used a previously determined TMV structure, EMPIAR-10021 [40]."

As stated in the response to Reviewer 1, point 2, we plan to include the comparison with the state-of-the-art data set on TMV. To clarify, the respective TMV images (EMPIAR-10305) were recorded at the EMBL Titan Krios microscope.

6. Analysis of the resolution and contribution of high spatial frequencies using layer line intensity ratios as a function of CSA is elegant and informative. It's also interesting that the STEM resolution in the reconstruction came out slightly better than the TEM for a comparable number of particles, implying that the information content per particle is higher in iDPC-STEM. This could be pursued a bit further by comparing reconstructions from subsets of the data. (an entirely optional suggestion)

Reviewer 1, point 7 asked for a similar comparison using the B-factor calculation via the Rosenthal-Henderson ResLog plot. Indeed, this analysis will be included in a revised version of the manuscript.

7. Depth sectioning with aberration correction should include reference to deJonge, for example <https://doi.org/10.1017/s1431927611012347>.

We thank the Reviewer for pointing out this comprehensive article on this topic and will include it in the revised version of the manuscript.

8. The paragraph in the discussion on aliasing (highlighted in yellow) describes a new result. It should better go into the results section, or maybe as an explanatory note in Supplementary Material. The bottom line is that the pixel spacing should fully sample the optical resolution, which is no surprise really. The semantics of cutoffs smaller or larger than Nyquist get confusing, and this is a distraction in the important discussion section.

We thank the Reviewer for this remark and will shorten the respective Discussion section for clarity and present this point in the Results section.

Reviewer #3:

Remarks to the Author:

1. Lazic and colleagues report the first high-resolution results from the application of iDPC-STEM (and STEM in general) to cryogenically frozen biological samples. This is an outstanding achievement and will be of great interest to researchers and manufacturers in the electron microscopy field. However, at this stage the technique does not appear to offer practical and theoretical advantages over the conventional cryo-EM approach, and therefore will be of limited interest to the broader structural biology community.

Reviewer 3 acknowledges the results as an outstanding achievement of great interest. At the same time, the Reviewer concludes that the technique lacks clear advantages. Therefore, we respectfully disagree with this conclusion of Reviewer 3. Reviewer 2 already pointed out that the practical advantages of cryo-iDPC-STEM over CTEM are limited (point 1a and b) although there are clear theoretical advantages of higher resolution and beam-induced motion. Therefore, we are aligned with Reviewer 2's opinion. We believe that Reviewer 3 proposes important aspects himself that are worth adding to the Discussion as they represent the principal benefits over CTEM imaging approaches.

In order to demonstrate the wider applicability of the technique, we plan to deliver an additional high-resolution data set on another sample than TMV (Reviewer 2, point 1c).

Several important aspects that could have made the work more attractive should have been discussed in detail:

1. In terms of performance, does iDPC-STEM have the potential to provide benefits over bright field cryo-EM? These may include higher resolution, higher throughput, access to more difficult samples, additional layers of quantitative information etc.

Based on the Reviewer's request, we plan to extend the Discussion section on potential future benefits. The outstanding results obtained in material science EM suggest that higher resolution, access to more difficult samples and quantitative information will also be provided in life science cryo-EM in the future.

2. In practical terms, what are the perspectives of iDPC-STEM for automation, ease of use, cost-effective and smaller footprint instrumentation etc.?

We started to discuss this point on page 5 of the manuscript. In the revised version of the manuscript, we will include an extended section on automation and ease of use in the Discussion. In line with our disclosed conflict-of-interest statement of some of the authors, we would like to refrain from a discussion regarding cost considerations and footprint of the instrumentation.

3. Overall, the authors should give an outlook on the future of the method and its potential advantages.

As requested, we will extend the outlook on the method to illustrate the potential advantages more clearly.

Decision Letter, second revision:

Dear Carsten,

It was very nice to talk to you today. Also, thank you for your letter asking us to reconsider our decision on your Article, "Single-particle cryo-EM structures from iDPC-STEM at near-atomic resolution". After careful consideration we have decided that we are willing to consider a revised version of your manuscript that that addresses the concerns about generalizability of the approach as well as makes a strong case for why biologist should consider this as an alternative to conventional approaches.

I am sure you understand that we cannot promise that we will send the paper back to the reviewers before seeing the revised manuscript.

- * include a point-by-point response to our referees and to any editorial suggestions
- * please underline/highlight any additions to the text or areas with other significant changes to facilitate review of the revised manuscript
- * address the points listed described below to conform to our open science requirements
- * ensure it complies with our general format requirements as set out in our guide to authors at www.nature.com/naturemethods
- * resubmit all the necessary files electronically by using the link below to access your home page

[REDACTED]

We hope to receive your revised paper within 3 months. If you cannot send it within this time, please let us know. In this event, we will still be happy to reconsider your paper at a later date so long as nothing similar has been accepted for publication at Nature Methods or published elsewhere.

OPEN SCIENCE REQUIREMENTS

REPORTING SUMMARY AND EDITORIAL POLICY CHECKLISTS

When revising your manuscript, please submit reporting summary and editorial policy checklists.

Please note that these forms are dynamic 'smart pdfs' and must therefore be downloaded and completed in Adobe Reader. We will then flatten them for ease of use by the reviewers. If you would like to reference the guidance text as you complete the template, please access these flattened

versions at <http://www.nature.com/authors/policies/availability.html>.

IMAGE INTEGRITY

DATA AVAILABILITY

Please include a "Data availability" subsection in the Online Methods. This section should inform readers about the availability of the data used to support the conclusions of your study, including accession codes to public repositories, references to source data that may be published alongside the paper, unique identifiers such as URLs to data repository entries, or data set DOIs, and any other statement about data availability. At a minimum, you should include the following statement: "The data that support the findings of this study are available from the corresponding author upon request", describing which data is available upon request and mentioning any restrictions on availability. If DOIs are provided, please include these in the Reference list (authors, title, publisher (repository name), identifier, year). For more guidance on how to write this section please see:

<http://www.nature.com/authors/policies/data/data-availability-statements-data-citations.pdf>

MATERIALS AVAILABILITY

SUPPLEMENTARY PROTOCOL

To help facilitate reproducibility and uptake of your method, we ask you to prepare a step-by-step Supplementary Protocol for the method described in this paper. We [ask you to prepare a step-by-step Supplementary Protocol for the method described in this paper. We](https://www.nature.com/nature-research/editorial-policies/reporting-standards#protocols)

[ask you to prepare a step-by-step Supplementary Protocol for the method described in this paper. We](https://www.nature.com/nature-research/editorial-policies/reporting-standards#protocols)

target="new">encourage authors to share their step-by-step experimental protocols on a protocol sharing platform of their choice and report the protocol DOI in the reference list. Nature Research's Protocol Exchange is a free-to-use and open resource for protocols; protocols deposited in Protocol Exchange are citable and can be linked from the published article. More details can be found at www.nature.com/protocolexchange/about.

ORCID

Nature Methods is committed to improving transparency in authorship. As part of our efforts in this direction, we are now requesting that all authors identified as 'corresponding author' on published papers create and link their Open Researcher and Contributor Identifier (ORCID) with their account on the Manuscript Tracking System (MTS), prior to acceptance. This applies to primary research papers only. ORCID helps the scientific community achieve unambiguous attribution of all scholarly contributions. You can create and link your ORCID from the home page of the MTS by clicking on 'Modify my Springer Nature account'. For more information please visit please visit www.springernature.com/orcid.

Sincerely,
Arunima

Arunima Singh, Ph.D.
Senior Editor
Nature Methods

Author Rebuttal, Second Revision:

Response (NMETH-A46924B-Z)

We appreciate the overall positive feedback on our cryo-iDPC-STEM manuscript. The three reviewers asked well-deserved questions in an effort to strengthen and widen the scope of the manuscript, which we fully endorse. In the response below, we addressed the raised items in a point-by-point fashion.

One of the main concerns was the general applicability of the cryo-iDPC-STEM technique. In order to demonstrate the wider applicability, we subjected another well-known single-particle specimen KLH to iDPC-STEM and resolved it to a resolution of 6.5 Å. The results are included in the revised version of the manuscript (Results and Figure 2). Other requests include additional data processing with an updated reference data set of TMV for better comparison, improved result presentation and extended discussions on the principal benefits of the imaging approach. These results are presented in new Figures S4 and S5 as well as updated Figures 4 and 5, S2 and S3.

Please find our detailed point-by-point response below.

Reviewer #1:

Remarks to the Author:

This work impressively documents that iDPC-STEM has to be considered as an alternative to CTEM data collection for frozen hydrated specimens. As such, this is an important milestone paper.

1. The main impact of this work will be with the community that applies Conventional TEM (CTEM) to frozen hydrated samples. This community is not very familiar with STEM. As this manuscript is the first significant contribution to the structural analysis of cryo-EM samples using STEM, the manuscript should be written so that the CTEM community can understand it.

According to the Reviewer's request, we made an extra-effort to describe the technical details of the STEM method for a wider audience in the revised manuscript. First, we complemented the Results section on page 4:

For a typical experiment with an opening angle of 2.0 mrad CSA (Fig. 1f), a probe spot of 4.9 Å effective diameter of the beam intensity moves over the sample line by line scanning every 2.4 Å spot resulting in an overlap of 50 % exposed area. Each spot exposure lasts a dwell time of 4 μs. The acquired spot signals are combined pixel by pixel to generate a complete 4096 x 4096 micrograph over a total of 68 s time yielding a field of view (FOV) of 983 nm.

Second, we included an additional paragraph and Figure S4/S5 on exact data acquisition setup in the Methods section in the revised manuscript (page 15, section iDPC-STEM imaging and acquisition):

For iDPC-STEM acquisition, the four-quadrant mode of the Panther STEM detection system (Thermo Fisher Scientific) was used. The detector has a circular layout with a hole in the center and is composed of 8 segments where each quarter circle is split into an inner and outer segment (Fig. S4a). The central hole has 1/6 inner radius in relationship to the overall detector dimension (Fig. S4b). The 8 segments are hard wired in such a way that radial separation is not used resulting in 4 quadrants with read-out capability. The beam size on the detector is set to cover more than 2/3 of the detector diameter such that the central hole radius corresponds less than 1/4 of the beam size (Fig. S4c). The scan grid geometry consists of a simple line-by-line layout contributing to each pixel of the reconstructed micrograph (Fig. S4d). The offset angle between the scanning coordinate frame and the detector coordinate frame, which determines the scan direction, is measured and set initially during the alignment. During data acquisition, the scan direction can be chosen freely and is taken into account by providing illumination center of mass vector components for the resulting scanning coordinate frame. This vector determination is critical to perform before the final integration step in iDPC-STEM takes place [7]. Note that only when the rotation angle of the scan direction is zero, the components of the DPC-

STEM vector can be simply determined by subtraction of the signal from the opposite quadrants [7, 8, 38].

2. It is not clear why the iDPC-STEM data were not compared with K2 data of Weis et al. (2019), which reached higher resolution. iDPC-STEM from today should be compared to CTEM from today, not from 6 years ago. The quantitative comparison of the strengths and weaknesses of iDPC-STEM could be done more rigorously.

We intentionally chose the older data set due to resolution comparability as the resolution is very close to the one obtained by the iDPC-STEM method. We fully agree, however, with the Reviewer's remark (see below Reviewer 2, point 5).

First, we included the data set and structure in the comparison panels of updated Figure 4 and Figure 5. We included both CTEM data sets and state in the Results section (page 7):

The latest CTEM data set from 2019 (EMPIAR-10305) went to 2.2 Å. When compared with the best iDPC-STEM map (4.0 mrad CSA) at 3.5 Å resolution, the earlier 2015 CTEM data set (EMPIAR-10021) went to slightly poorer resolution of 3.7 Å. Local resolution assessment corroborates the better map quality of the 4.0 mrad CSA iDPC-STEM structure in comparison with the 2015 CTEM structure albeit worse than the 2019 CTEM structure (Fig. S3c).

Second, for a more rigorous comparison, we included the B-factor estimation using the Rosenthal-Henderson plot with the CTEM data set for KLH and TMV (Figure S3c) and describe this analysis in the Results section (page 8):

B-factor estimations that assess the data set quality as a whole involve the analysis of the particle as a function of the obtained resolution [47]. Using particle subsets of CTEM (EMPIAR-10021), CTEM (EMPIAR-10305), 3.5 mrad iDPC-STEM KLH and 4.0 mrad iDPC-STEM TMV micrographs, we determined B-factors by logarithmic regression to 147, 62, 437 and 93 Å², respectively (Fig. S3d). A global B-factor of 62 Å² confirms that the present iDPC-STEM micrographs are of better quality than the CTEM acquisition from 2015 recorded employing a Falcon II DED [42], and of worse quality than the 2019 CTEM acquisition by a K2 DED CTEM acquisition in 2019 [34].

3. iDPC-STEM scans the electron beam in a 700nm long line over the sample, before returning to collect the next scan line. (Does the beam go back-and-forth over the sample, or forth-and-forth, i.e., the return is without significant beam exposure?)

As requested, we detailed the exact line-by-line geometry sequence of the scan in a new paragraph of the abovementioned Results and Methods section supported by the new Figure S4d (Reviewer 1, point 1).

4. In STEM mode, any protein of a few nm dimensions will therefore be exposed by the electron beam in several scan lines that are separated by time. Structural STEM data are collected of a protein "after" the adjacent part of the protein has already been fully beam-damaged by 35 e/Å². In contrast, in CTEM mode, data are collected "while" beam damage occurs, and importantly the

first frames of a dose-fractionated movie-mode real-space imaging collects data on the entire protein that is still in a non-beam damaged state. This in the viscous environment of vitrified water may or may not make a difference to the SNR vs. electron beam damage of recorded data. This manuscript could discuss this fundamental difference between the two illumination modes.

As suggested by the Reviewer, we elaborated on this critical difference between the imaging strategies of CTEM vs STEM in the Discussion section of the revised manuscript on page 11:

To compare CTEM and STEM acquisitions, we emphasize their fundamental difference with respect to electron delivery and energy deposition. Unlike with flood-beam illumination in CTEM where electrons are delivered everywhere at once, the scanning mode deposits electrons sequentially one pixel position at a time. A potential benefit of the STEM approach may be that the energy transferred to the sample can spread and dissipate towards non-illuminated areas, presumably weakening the impact and the damage at the exposed spot. Due to the overlapping geometry of the scan, i.e. effective beam size is larger than a pixel, the sample spots are exposed multiple times and, consequently, the total dose per pixel is accumulating in STEM. In contrast, CTEM exposed areas are illuminated once and, in some cases, damage-free maps can even be extrapolated at zero electron exposure [56]. In analogy to STEM, spot scanning of 100 nm beams was implemented in CTEM and was shown to mitigate beam-induced motion and to improve contrast for vitrified specimens [57]. The STEM approach offers additional opportunities for evaluating different electron delivery strategies by e.g. changing the scan grid order. This scanning scheme may give rise to reduced radiation damage effects when compared with typical flood-beam illumination approaches. A series of studies investigating the damage mechanisms in STEM has been reported in 2D materials using alternative scan patterns, which indicate that they may further reduce beam damage [58, 59, 60]. The typical CTEM exposure causes beam-induced motion and leads to ice-patch or particle movements and ultimately to image blur, which is now commonly compensated by motion correction [61]. In this manuscript, motion correction was also applied to the 2015 and 2019 TMV CTEM reference data sets. For STEM, the locally induced beam motion building up throughout the scan may result in image distortions and anisotropic magnifications in the reconstructed micrograph. Nevertheless, without the employment of any correction strategies, the present iDPC-STEM experiments show that a simple grid scanning scheme can be used to generate near-atomic resolution structures of TMV.

5. In CTEM, the accumulating beam damage can be quantified in B-factor curves for each movie frame. In this manuscript, the authors mention the possibility of dose-fractionated iDPC-STEM, by which the authors presumably mean to perform several STEM scans of the same sample region (line 305). The authors could discuss the time requirements for such multiple-scan STEM approach.

In one sentence of the Discussion (page 12), we already alluded to the future possibility of employing a motion correction type during the STEM acquisition:

Additionally, recording individual frames in movie mode will reduce the effects of beam-induced motion and further improve the image quality of cryo-iDPC-STEM.

The technical implementation of such a feature can only be superficially sketched and contains uncertainty at this point in time. Critical for such an implementation will be the adaption of the

scan engine with respect to hardware and software to enable shorter dwell times. Currently dwell times for each scan of several μs are being used (Table 4). These dwell times should be reduced by at least an order of magnitude to allow for a movement correction.

6. In CTEM, sample drift over time or over electron dose affects collected data in form of particle movements, which are tracked and corrected by image processing of dose-fractionated movies. In STEM, sample drift over time results in image distortions such as skewing or anisotropic magnification, and sample drift over electron dose will result in resolution loss of projected particle images. The authors report that their STEM data do not appear to suffer critically from beam-induced movement. How do they quantify this? Do datasets recorded at different electron fluences show different features in terms of Guinier B-factors or particle distortions or resolution losses? This could also be discussed.

We agree with the Reviewer that more care should be taken to delineate the different effects of image distortions in CTEM vs. STEM imaging. Therefore, we analyzed possible magnification differences by Fourier analysis of layer lines using differently aligned TMV rods.

When taking into account the $1/23$ and $1/(11.5 \text{ \AA})$ layer line, no significant magnification deviations in different x-y directions could be identified (Figure below): Three stacked collapsed layer line profiles from exemplary micrographs with TMV rods aligned almost horizontally (left) and aligned almost vertically (right). Observed minor deviations can also be a result of out-of-plane tilt of the TMV rods.

7. In CTEM, the performance of the entire imaging chain (consisting of electron source, lenses, sample identity and quality, and detector) can be quantified by the ResLog B-factor calculated from final data sets. Conventionally, instrument performances in CTEM are compared with each other when using apoferritin as test sample. This ResLog B-factor reports on the SNR of recorded particles. This is different than the Guinier B-factor that reports on the amplitude decay towards higher resolution.

Ideally, the iDPC-STEM method should be compared to CTEM by establishing a measure of the ResLog B-factor of this method when applied to apoferritin. If the authors are not able to perform that in the scope of this manuscript, then they should at least calculate a ResLog B-factor for TMV from a comparable instrument (with a current dataset, not one from 2015) and compare that with a ResLog B-factor for these iDPC-STEM data.

As requested by the Reviewer, we performed the B-factor calculation (Reviewer 1, point 2b) and added the Rosenthal-Henderson ResLog plot with our STEM KLH 3.5 mrad, STEM TMV 4.0 mrad, CTEM TMV data sets in the Results section (page 8) the revised version of the manuscript (FigureS3d).

8. In CTEM, contrast is maximized by optimizing the defocus setting, whereby the operator attempts to balance the phase contrast vs. resolution of the instrument's CTF.

In iDPC-STEM as applied here, the tool records a significant amount of phase contrast. The CTF of iDPC-STEM thereby sensitively depends on the defocus setting. With a 50nm thick sample protein sample, a particle at the top and at the bottom surface of the ice layer should have a significantly different CTF, which is later not fitted in Relion and which therefore should limit resolution. In contrast to CTEM, where defocus is later fitted during image processing and where the defocus difference between top and bottom parts of the same particle are corrected through Ewald sphere curvature correction, accurate focusing of the iDPC-STEM during data collection is important for reaching high resolution. This should be discussed here.

We thank the Reviewer for bringing up this topic as it arises from a misunderstanding. In the thickness-of-the-sample treatment of the Discussion (page 10), the key parameter to be considered is the depth of focus of the beam. In the CSAs used throughout the manuscript, the depth of focus is significantly larger than the sample thickness, therefore CTF variations within the sample do not need to be considered. We added a clarifying statement to the relevant manuscript section (page 10):

This property of STEM imaging is substantially different to CTEM acquisitions when different particle z-positions within the ice layer give rise to different projections due to the associated change in CTF.

9. How does the phase contrast CTF of a CTEM compare with the phase contrast CTF of iDPC-STEM? It is clear that iDPC-STEM CTF curves are sample dependent and dependent on detector dimensions and defocus, so that an objective comparison of the two CTF curves is difficult to provide. However, as this is the main message of this manuscript, the CTF performance of the two should at least be discussed quantitatively, including a discussion of the SNR vs. beam damage at higher resolution between the two methods.

As requested by the Reviewer, we added a comparative graph on the typical CTEM-CTF including phase contrast employed in the reference TMV data sets (new Figure S2e and S2f) in addition to the existing STEM-CTF (former Figure S2e, updated Figure S2g) in the Results section (page 4):

In comparison with an oscillating CTF of a CTEM image acquired in underfocus, the signal in iDPC-STEM further decays almost linearly with higher resolutions, reaching zero at the theoretical resolution limit of the given CSA (Fig. S2d-S2g).

For further reference, the iDPC-STEM CTFs are well understood, analytically expressed and explained in depth [Lazic and Bosch 2017; DOI: 10.1016/bs.aiep.2017.01.006]. In addition, SNR considerations of iDPC-STEM in comparison with CTEM have been treated in detail in the following manuscript [Lazic et al., 2016; DOI: 10.1016/j.ultramic.2015.10.011]. Therefore, we included a discussion on the comparison between damage mechanisms of STEM vs CTEM in the response to Reviewer 1, point 4.

10. According to TFS documentation, the Panther STEM detector consists of two different detectors, one with eight quadrants surrounding a central hole, and one with eight quadrants without hole.

This information is now given in the extended Methods section in response to point 1 of Reviewer 1.

11. What was the scan direction with respect to the orientation of the four-quadrant detector?

This information is now given in the extended Methods section in response to point 1 of Reviewer 1.

12. This manuscript suggests that the detector with the central hole was used here. Is that so? On the other side, you write in line 224 that iDPC-STEM uses all the available electrons, suggesting that you used the detector without the hole. Which one is it?

If the one with the hole was used: At the different illumination convergence semi angles used, what were the inner and outer radii in mrad of the inner and outer quadrant segments of the Panther STEM detector used here? Which quadrants were used, the four inner or the four outer segments, or the four combinations of inner and outer segments? How was the signal from the quadrants evaluated to produce the signal, i.e., how does the system calculate the center of mass movement? How sensitive is this when single electron counting with very few electrons is applied?

iDPC-STEM uses all signal-relevant electrons. The central hole radius is kept small enough to ensure that no effect on the CTF is imposed (please see [8], Sec. 4.4). This information is now given in the indicated line and in the extended Methods section in response to point 1 of Reviewer 1.

Reviewer #2:

Remarks to the Author:

The manuscript by Lazic et al demonstrates that iDPC-STEM can be a useful tool in cryo-EM to determine the structure of biomolecules at high resolution. Quoting from the paper, "This study shows that iDPC-STEM imaging can be successfully applied to cryo-EM single-particle based structure determination and elucidate biological structures at near-atomic resolution." There is a long-standing notion in life science EM that STEM is not useful for cryo-imaging, while the materials scientists have been charging ahead with cryo-STEM and have reached spectacular resolutions on sensitive materials with very low dose. Until now cryo-STEM in life science was applied mainly to tomography of thick samples, where cryo-TEM fails due to inelastic scattering. Expansion into high resolution on molecular specimens depends on phase contrast, which is implemented in the iDPC technique developed by Lazic and co-workers at the FEI/TFS research labs. According to the contributions list he collected the data himself at the lab of Sachse in Juelich that is very experienced in single particle analysis and especially in TMV studies. This seems a good recipe for success. iDPC has many known advantages that are exploited in materials science, even if it is not yet entirely routine practice. Known advantages include the visibility of heavy elements next to light ones (unlike HAADF) and the availability of sub-atomic resolution at very low dose. Complete contrast transfer is also known, at least theoretically, because the images are acquired in focus. The current manuscript demonstrates that very impressively. The absence of beam-induced sample movement is a new result as far as I'm aware. That is very encouraging for the further development of STEM methods for cryo-EM. On technical grounds the work is a major step forward.

We appreciate genuine acknowledgment and encouragement by the reviewer of our cryo-STEM attempt presented in this manuscript. Please find our in-detail responses to the remarks and comments below.

1. a) As a Nature Methods paper I'm ambivalent. It's not clear from the work what advantages iDPC-STEM holds over the current state of the art. Why would one choose that method instead of existing TEM in order to get the best data? The CTF contrast inversions in TEM are annoying, but they are corrected very successfully by standard image processing pipelines. Is the iDPC method potentially faster, or cheaper, or easier than conventional TEM?

b) It does not show that iDPC-STEM would be a method of choice for someone who already has a Krios-level microscope and DED camera, nor would it convince someone interested in molecular structure "as fast as possible" to invest in iDPC rather than the best TEM camera. The manuscript leaves me wondering, what can it do that TEM cannot? The conclusions are sound, but more scientific than practical. That stated, publication in Nature Methods is a decision for the editors. More technical comments follow.

The Reviewer acknowledges that the presented work represents a major technical step forward. At the same time, the Reviewer raises a number of highly interesting points regarding the practical consequences, e.g. whether iDPC-STEM is faster, cheaper and easier than CTEM or even the method of choice between the two. We feel that this point cannot be easily addressed within the scope of this manuscript as it would require an entirely new study including optimized hardware design with speed and ease of use in mind. We feel that we cannot compare the cryo-CTEM level of maturity and productivity including the current speed and ease of use at this point in time considering the amount of time (~30 years) and efforts that went into perfecting cryo-CTEM operations. We believe, however, that this manuscript may mark the beginning of such a development as it demonstrates the principal applicability of the technique.

c) Can it be applied more widely? In fact TMV is a very particular specimen. All the particles lie flat and almost perfectly ordered along the grid. There is no concern about preferential orientations or slight differences in defocus per particle. In this way TMV is a relatively easy case. The manuscript shows very nicely that iDPC-STEM produces high resolution images that are amenable to single particle processing for TMV.

TMV is indeed a very good test specimen for cryo-EM imaging in particular, and structural biology in general. In order to demonstrate wider applicability to other specimens, we extended the analysis on the second test specimen KLH. We had already presented KLH in Figure 1 as a micrograph example and now we included a newly acquired data set and the 3D reconstruction. We added a new paragraph in the Results section (page 5) and a new Figure 2. The main result is a 6.5 Å cryo-EM structure of KLH:

In order to quantitatively analyze larger numbers of KLH images, we collected a total of 760 micrographs at CSA of 2.0 mrad. After extracting a total of 9,150 particles and performing routine 2D classification procedures [42], we identified a series of averaged characteristic side, top and skewed side views of KLH compatible with D5 symmetry (Fig. 2a). In order to improve resolution for a 3D reconstruction, we collected additional 687 micrographs using a larger CSA of 3.5 mrad, extracted 18,597 particles and determined the structure of KLH with D5 symmetry imposed at 6.5/6.8 Å (0.143 and FDR-FSC criterion) (Fig. 2b, c), which is beyond the latest reported resolution [43]. We docked the available structure PDB-ID 4BED into the density and found good agreement with the expected secondary structural elements at the determined resolution (Fig. 2d). These data

show that iDPC-STEM micrographs of a single-particle specimen can be used to determine the sub-nanometer molecular structure of KLH. Further improvements in resolution of the 3D structure of KLH would require a data set size orders of magnitude larger. In the light of this consideration, we turned to the molecular specimen TMV, containing a high number of asymmetric units per unit length, which is ideally suited for efficient structural averaging using a smaller number of micrographs.

3. The iDPC detector is not standard equipment, at least not in life science EM. It deserves a more complete description than the single sentence on page 12. A drawing would be useful, or at least a link to a full description. The heart of the new method is left as a commercial black box.

As requested by the Reviewers, we included the detector/scanning scheme drawing in the Methods section and more detailed description on scanning and detection/acquisition process (Reviewer 1, point 1).

3. Fig 2 is very impressive. Please clarify if the images were processed in any way, for example background correction or high pass filtering. The texture differs from the panels of Fig 3 on the left, and looks more like the filtered images under the heading Segmented Particles on the right.

We agree with the Reviewer that the preprocessing of the iDPC-STEM images has a significant impact on the appearance and visual interpretability, predominantly for the low spatial frequencies. Although it was mentioned in the Methods section, we highlight this critical point by adding the following statement to the Results (page 5):

For consistent interpretation of iDPC-STEM images, we apply a high-pass filter at 251 Å FWHM referring to them as preprocessed iDPC-STEM images.

We also matched the grey scale of the image insets (left panels) of former Figure 3 (now Figure 4) visually to avoid misinterpretation. In the legends of Figures 1, 2 and 3, S5, to avoid any confusions we now specify “preprocessed micrographs” when appropriate.

4. The data were acquired on a Krios microscope that has a Falcon 4 DED camera installed. It was used for screening. A side by side comparison of the iDPC-STEM with the Falcon images at high magnification would be useful and convincing. It would also be interesting to see a comparison with conventional HAADF images (if those are already available).

As requested by Reviewer 1, point 2, we updated the manuscript by an in-depth side-by-side comparison of the iDPC-STEM images and deposited TMV data set (EMPIAR-10305) taken with the latest acquisition procedures and detector (check Figures 4 and 5). Simultaneous imaging in STEM is another advantage over CTEM that we describe in the revised manuscript (page 15, Figure S5b,c):

With an additional ADF detector, simultaneous acquisition of ADF/ABF and iDPC-STEM data is possible. Exemplary micrographs of vitrified densely packed TMV rods confirmed improved contrast and high-resolution information transfer of iDPC-STEM acquisitions (Fig. S5b,c).

5. It is strange that the TEM benchmark was the dataset EMPIAR-10021 from 2015 in Ref 40, rather than the more recent study of the same group in 2019 (Ref 33, EMPIAR-10305) at 1.9Å. This is the current state of the art. Most likely the data also come from the same microscope since

the correspondence address is in Juelich rather than EMBL. The only comment in the text is "As a defocused-based CTEM reference data set, we used a previously determined TMV structure, EMPIAR-10021 [40]."

We addressed this concern by the response to Reviewer 1, point 2.

6. Analysis of the resolution and contribution of high spatial frequencies using layer line intensity ratios as a function of CSA is elegant and informative. It's also interesting that the STEM resolution in the reconstruction came out slightly better than the TEM for a comparable number of particles, implying that the information content per particle is higher in iDPC-STEM. This could be pursued a bit further by comparing reconstructions from subsets of the data. (an entirely optional suggestion)

We addressed this item in Reviewer 1, point 2b and 7.

7. Depth sectioning with aberration correction should include reference to deJonge, for example <https://doi.org/10.1017/s1431927611012347>.

We thank the Reviewer for pointing out this comprehensive article on this topic and included it in the Discussion of the revised version of the manuscript (page 10).

8. The paragraph in the discussion on aliasing (highlighted in yellow) describes a new result. It should better go into the results section, or maybe as an explanatory note in Supplementary Material. The bottom line is that the pixel spacing should fully sample the optical resolution, which is no surprise really. The semantics of cutoffs smaller or larger than Nyquist get confusing, and this is a distraction in the important discussion section.

We thank the Reviewer for this remark and shortened the respective Discussion section for clarity and present this point in the Results section and converted Supplementary Figure to Figure 6.

Reviewer #3:

Remarks to the Author:

1. Lazic and colleagues report the first high-resolution results from the application of iDPC-STEM (and STEM in general) to cryogenically frozen biological samples. This is an outstanding achievement and will be of great interest to researchers and manufacturers in the electron microscopy field. However, at this stage the technique does not appear to offer practical and theoretical advantages over the conventional cryo-EM approach, and therefore will be of limited interest to the broader structural biology community.

Reviewer 3 acknowledges the results as an outstanding achievement of great interest. Reviewer 2 had a similar remark, pointing out that the practical advantages of cryo-iDPC-STEM over CTEM are at the moment still limited (point 1a and b) despite the theoretical advantages of higher resolution and beam-induced motion. We are in alignment with Reviewer 2's opinion in this respect, and we responded accordingly. We believe that Reviewer 3 still proposes important aspects that we consider worth adding to the Discussion as they represent the principal benefits over CTEM imaging approaches. Find the response to the detailed points below.

Several important aspects that could have made the work more attractive should have been discussed in detail:

1. In terms of performance, does iDPC-STEM have the potential to provide benefits over bright field cryo-EM? These may include higher resolution, higher throughput, access to more difficult samples, additional layers of quantitative information etc.

Based on the Reviewer's request, we extended the Discussion section on potential future benefits (page 13):

In this study, we demonstrated the principal applicability of STEM methods to cryo-vitrified biological specimens for 3D structure determination. Advanced STEM methods or derivatives have been used in materials science for resolving sub-50 pm spatial detail [4, 5]. Some of these methods, like iDPC-STEM [7, 8], have been shown to be very dose efficient and, therefore, have been applied successfully to dose-sensitive specimens [18, 19]. Images can be obtained with highest contrast in focus without the typical contrast inversions at higher frequencies known from CTEM. One of the further benefits of iDPC-STEM over CTEM is the improvement in low-resolution contrast when employing smaller CSAs to target intermediate resolutions. Therefore, largest benefits can be expected when no or little averaging of cryo-images is possible. In the case of biological vitrified samples, this feature may be particularly useful for the visualization of complex environments such as cellular lysates and biological cells [20]. In CTEM, the thickness of the specimen imposes a significant limitation on such samples as contrast blurs due to incoherency and inelastic scattering. The problem remains critical even when in-focus images are taken with a phase plate. For STEM, the contrast will not be significantly deteriorated by inelastic scattering, also at lower voltages the image remains sharp and shows full contrast even when the object thickness becomes comparable to the inelastic mean free path [68]. Dynamic focusing across a scan field of a tilted specimen is possible and could provide benefits in single-particle analysis in order to overcome preferred particle orientations. Once iDPC-STEM imaging can be applied to tilt series of thicker biological specimens, cryo-electron tomographic reconstructions will benefit from the complete contrast transfer in STEM images devoid of any oscillating CTF while still preserving high-resolution information. Moreover, alternative STEM acquisition schemes [60] and micrograph reconstruction approaches [27], when further developed, should also prove beneficial for imaging biological samples. The STEM approach opens up additional opportunities for correlative sample characterization, e.g. spectroscopic mapping by energy dispersive X-ray (EDX) spectroscopy or electron energy loss spectroscopy (EELS) to analyze the element composition of a specimen.

2. In practical terms, what are the perspectives of iDPC-STEM for automation, ease of use, cost-effective and smaller footprint instrumentation etc.?

We started to address this point in the Discussion (page 12) of the manuscript up to a point that we found appropriate. In line with our disclosed conflict-of-interest statement of some of the authors, we prefer to focus on the scientific rather than economic aspects and considerations of the technique.

Automation of the STEM acquisition procedures, however, will be straightforward to employ in the future. Detectors used for iDPC-STEM allow scan speeds of two orders of magnitude faster than the ones used in this work, which will ultimately reduce acquisition times to below one second. This way, when hardware improvement supported by software-based automation is further implemented, it will become possible to image

equally large data sets of single-particle specimens in similar time frames as in CTEM. Additionally, recording individual frames in movie mode will reduce the effects of beam-induced motion and further improve the image quality of cryo-iDPC-STEM.

3. Overall, the authors should give an outlook on the future of the method and its potential advantages.

We addressed this item in Reviewer 3, point 1.

Decision Letter, third revision:

Dear Carsten,

Thank you for submitting your revised manuscript "Single-particle cryo-EM structures from iDPC-STEM at near-atomic resolution" (NMETH-A46924C). It has now been seen by the original referees and their comments are below. The reviewers find that the paper has improved in revision, and therefore we'll be happy in principle to publish it in Nature Methods, pending minor revisions to satisfy the referees' final requests and to comply with our editorial and formatting guidelines.

TRANSPARENT PEER REVIEW

Thank you again for your interest in Nature Methods Please do not hesitate to contact me if you have any questions.

Sincerely,
Arunima

Arunima Singh, Ph.D.

Senior Editor
Nature Methods

ORCID

Reviewer #1 (Remarks to the Author):

Lazik et al. present the revised version of the manuscript NMETH-A46924A-Z. This manuscript is now a very interesting and important milestone paper on the path towards translating the recent developments in materials sciences STEM to life sciences cryo-EM specimen. My earlier comments were addressed to my fullest satisfaction.

The manuscript provides a very interesting quantitative comparison of cryo-STEM with cryo-CTEM. The work allows a side-by-side comparison of difficulties and promises of each method, now comparing apples with apples in a clearly documented manner.

A few minor details remain from my side that merit attention:

Figure S2f shows the CTF of a CTEM at two different defocus settings, which includes the incoherence envelope. The spatial envelope function could be included as well, as that is defocus dependent and then would show that at higher defocus the envelope function decays faster, showing the dilemma in CTEM of life sciences specimens of having to choose between higher defocus for contrast and lower defocus for a better envelope.

See for example equation [28.53] on page 471 of Williams et al., "TEM", Springer (1996), available online here:

https://link.springer.com/content/pdf/10.1007/978-1-4757-2519-3_28.pdf

Figure S3d shows the reslog B-factor plots of TMV and KLH. I recommend to either split TMV and KLH into two different panels, or label the two different specimens more clearly. For the first three lines, it would be useful to add "TMV" to the label, such as:

x TMV: CTEM (2015, EMPIAR-10021): Reslog B-factor = $147 \pm 25 \text{ \AA}^2$

x TMV: CTEM (2019, EMPIAR-10305): Reslog B-factor = $62 \pm 4 \text{ \AA}^2$

x TMV: iDPC-STEM (4.0 mrad): Reslog B-factor = $93 \pm 9 \text{ \AA}^2$

x KLH: iDPC-STEM (2.0 mrad): Reslog B-factor = $437 \pm 40 \text{ \AA}^2$

It would also be helpful to indicate on the vertical axis the resolution in \AA that different values represent. 0.025 corresponds to 6.32 \AA , 0.200 corresponds to 2.24 \AA .

Reviewer #2 (Remarks to the Author):

The manuscript is greatly improved by the revisions. The concerns of all three referees were very

similar and to my reading they have been addressed effectively. As a technical advance, the work is indeed an important milestone. As a methods paper, the instrumentation and protocols have now been described at a level that should permit replication. I have no further concerns in that regard.

As a "Nature Methods" paper there is an additional factor on which the editors will have to adjudicate. No matter how great a step forward, the method as presented does not represent an advance over the current state of the art for the structural biologist. CTEM is working so well for macromolecular structure that improvement is a tall order. After reading the paper, any practitioner aiming to solve a new structure would still turn to CTEM rather than iDPC-STEM. The KLH result is an improvement over existing structures in the EMDB, but those were obtained in the CCD era prior to the resolution revolution. The discussion describes a number of interesting extensions where the iDPC-STEM approach has potentially significant advantages but these were not demonstrated. For all the effort, this paper may be cited simply as a demonstration that CTEM is superior to STEM for cryo-EM. I doubt that this was the authors intent.

Reviewer #3 (Remarks to the Author):

The authors have addressed adequately the main points.
- Line 235, "A global B-factor of 62 Å² ..." -> 93 Å²

Author rebuttal, third revision

Response (NMETH-A46924D)

We appreciate the positive feedback on our cryo-iDPC-STEM manuscript. The three reviewers pointed out minor remaining items. In the response below, we addressed the raised items in a point-by-point fashion.

Reviewer #1:

Remarks to the Author:

Lazik et al. present the revised version of the manuscript NMETH-A46924A-Z.

This manuscript is now a very interesting and important milestone paper on the path towards translating the recent developments in materials sciences STEM to life sciences cryo-EM specimen. My earlier comments were addressed to my fullest satisfaction.

The manuscript provides a very interesting quantitative comparison of cryo-STEM with cryo-CTEM. The work allows a side-by-side comparison of difficulties and promises of each method, now comparing apples with apples in a clearly documented manner.

A few minor details remain from my side that merit attention:

Figure S2f shows the CTF of a CTEM at two different defocus settings, which includes the incoherence envelope. The spatial envelope function could be included as well, as that is defocus dependent and then

would show that at higher defocus the envelope function decays faster, showing the dilemma in CTEM of life sciences specimens of having to choose between higher defocus for contrast and lower defocus for a better envelope.

See for example equation [28.53] on page 471 of Williams et al., "TEM", Springer (1996), available online here:

https://link.springer.com/content/pdf/10.1007/978-1-4757-2519-3_28.pdf

According to the Reviewer's request, we updated the Figure S2f and included an incoherence envelope into the two different defocus settings.

Figure S3d shows the reslog B-factor plots of TMV and KLH. I recommend to either split TMV and KLH into two different panels, or label the two different specimens more clearly. For the first three lines, it would be useful to add "TMV" to the label, such as:

x TMV: CTEM (2015, EMPIAR-10021): Reslog B-factor = $147 \pm 25 \text{ \AA}^2$

x TMV: CTEM (2019, EMPIAR-10305): Reslog B-factor = $62 \pm 4 \text{ \AA}^2$

x TMV: iDPC-STEM (4.0 mrad): Reslog B-factor = $93 \pm 9 \text{ \AA}^2$

x KLH: iDPC-STEM (2.0 mrad): Reslog B-factor = $437 \pm 40 \text{ \AA}^2$

It would also be helpful to indicate on the vertical axis the resolution in \AA that different values represent. 0.025 corresponds to 6.32 \AA , 0.200 corresponds to 2.24 \AA .

In line with the Reviewer's request, we updated Figure S3d panel with the amended legend text as well as an additional label axis.

Reviewer #2:

The manuscript is greatly improved by the revisions. The concerns of all three referees were very similar and to my reading they have been addressed effectively. As a technical advance, the work is indeed an important milestone. As a methods paper, the instrumentation and protocols have now been described at a level that should permit replication. I have no further concerns in that regard.

Reviewer #3:

Remarks to the Author:

The authors have addressed adequately the main points.

- Line 235, "A global B-factor of 62 \AA^2 ..." -> 93 \AA^2

We corrected the typo as requested.

Final Decision Letter:

Dear Carsten,

I am pleased to inform you that your Article, "Single-particle cryo-EM structures from iDPC-STEM at near-atomic resolution", has now been accepted for publication in Nature Methods. Your paper is tentatively scheduled for publication in our September print issue, and will be published online prior to that. The received and accepted dates will be 24th August 2021 and 19th July 2022. This note is intended to let you know what to expect from us over the next month or so, and to let you know where to address any further questions.

Please note that *Nature Methods* is a Transformative Journal (TJ). Authors may publish their research with us through the traditional subscription access route or make their paper immediately open access through payment of an article-processing charge (APC). Authors will not be required to make a final decision about access to their article until it has been accepted. Find out more about Transformative Journals

Your paper will now be copyedited to ensure that it conforms to Nature Methods style. Once proofs are generated, they will be sent to you electronically and you will be asked to send a corrected version within 24 hours. It is extremely important that you let us know now whether you will be difficult to contact over the next month. If this is the case, we ask that you send us the contact information (email, phone and fax) of someone who will be able to check the proofs and deal with any last-minute problems.

If, when you receive your proof, you cannot meet the deadline, please inform us at

rjsproduction@springernature.com immediately.

Once your manuscript is typeset and you have completed the appropriate grant of rights, you will receive a link to your electronic proof via email with a request to make any corrections within 48 hours. If, when you receive your proof, you cannot meet this deadline, please inform us at rjsproduction@springernature.com immediately.

Once your paper has been scheduled for online publication, the Nature press office will be in touch to confirm the details.

Once your paper has been scheduled for online publication, the Nature press office will be in touch to confirm the details.

Content is published online weekly on Mondays and Thursdays, and the embargo is set at 16:00 London time (GMT)/11:00 am US Eastern time (EST) on the day of publication. If you need to know the exact publication date or when the news embargo will be lifted, please contact our press office after you have submitted your proof corrections. Now is the time to inform your Public Relations or Press Office about your paper, as they might be interested in promoting its publication. This will allow them time to prepare an accurate and satisfactory press release. Include your manuscript tracking number NMETH-A46924D and the name of the journal, which they will need when they contact our office.

About one week before your paper is published online, we shall be distributing a press release to news organizations worldwide, which may include details of your work. We are happy for your institution or funding agency to prepare its own press release, but it must mention the embargo date and Nature Methods. Our Press Office will contact you closer to the time of publication, but if you or your Press Office have any inquiries in the meantime, please contact press@nature.com.

Nature Research journals encourage authors to share their step-by-step experimental protocols on a protocol sharing platform of their choice. Nature Research's Protocol Exchange is a free-to-use and open resource for protocols; protocols deposited in Protocol Exchange are citable and can be linked from the published article. More details can be found at www.nature.com/protocolexchange/about.

Please note that you and any of your coauthors will be able to order reprints and single copies of the

issue containing your article through Nature Research Group's reprint website, which is located at <http://www.nature.com/reprints/author-reprints.html>. If there are any questions about reprints please send an email to author-reprints@nature.com and someone will assist you.

Best regards,
Arunima

Arunima Singh, Ph.D.
Senior Editor
Nature Methods